# An integrative approach to protein sequence design through multiobjective optimization

Lu Hong[1]*, Tanja Kortemme[1,2,3]*

**1** Department of Bioengineering and Therapeutic Sciences, University of California, San Francisco, California, United States of America, **2** Quantitative Biosciences Institute, University of California, San Francisco, California, United States of America, **3** Chan Zuckerberg Biohub, San Francisco, California, United States of America

\* honglu88@gmail.com (LH); tanjakortemme@gmail.com (TK)

## Abstract

With recent methodological advances in the field of computational protein design, in particular those based on deep learning, there is an increasing need for frameworks that allow for coherent, direct integration of different models and objective functions into the generative design process. Here we demonstrate how evolutionary multiobjective optimization techniques can be adapted to provide such an approach. With the established Non-dominated Sorting Genetic Algorithm II (NSGA-II) as the optimization framework, we use AlphaFold2 and ProteinMPNN confidence metrics to define the objective space, and a mutation operator composed of ESM-1v and ProteinMPNN to rank and then redesign the least favorable positions. Using the two-state design problem of the foldswitching protein RfaH as an in-depth case study, and PapD and calmodulin as examples of higher-dimensional design problems, we show that the evolutionary multiobjective optimization approach leads to significant reduction in the bias and variance in RfaH native sequence recovery, compared to a direct application of ProteinMPNN. We suggest that this improvement is due to three factors: (i) the use of an informative mutation operator that accelerates the sequence space exploration, (ii) the parallel, iterative design process inherent to the genetic algorithm that improves upon the ProteinMPNN autoregressive sequence decoding scheme, and (iii) the explicit approximation of the Pareto front that leads to optimal design candidates representing diverse tradeoff conditions. We anticipate this approach to be readily adaptable to different models and broadly relevant for protein design tasks with complex specifications.

## Author summary

Proteins are the fundamental building blocks of life, and engineering them has broad applications in medicine and biotechnology. Computational methods that seek to model and predict the protein sequence-structure-function relationship have seen significant advancement from the recent development in deep learning. As more models become available, it remains an open question how to effectively combine them into a coherent computational design approach. One approach is to perform computational design with one model, and filter the design candidates with the others. In this work, we demonstrate

**Data Availability Statement:** All code for methods described in this work and for generating and analyzing the benchmark data can be accessed at https://github.com/luhong88/int_seq_des.

**Funding:** This work was supported by a grant from the National Institutes of Health (R35 GM145236 to T.K.). T.K. is a Chan Zuckerberg Biohub Investigator. The funders had no role in study design, data collection and analysis, decision to publish, or preparation of the manuscript.

**Competing interests:** The authors have declared that no competing interests exist.

how an optimization algorithm inspired by evolution can be adapted to provide an alternative framework that outperforms the post hoc filtering approach, especially for problems with multiple competing design specifications. Such a framework may enable researchers to more effectively integrate the strengths of different modeling approaches to tackle more complex design problems.

## Introduction

The field of computational protein design has achieved major breakthroughs in recent years [1] in terms of its ability to design proteins and protein assemblies with diverse folds and functions (see, e.g., [2–9]), which has already found application in the design of therapeutically relevant biomolecules such as vaccines [10] and antibodies [11,12]. Such breakthroughs are built upon improvements in atomistic modeling techniques, such as the Rosetta software suite [13], and recent advances in machine learning-based structure prediction models [14–18], sequence design (or inverse folding) models [19–25], protein language models [26–34], and denoising diffusion probabilistic models [35–44].

As more models become available and the space of designable protein machinery becomes more sophisticated, there is an increasing need for frameworks that can integrate multiple models and objective functions directly into the protein design process, in order to take advantage of the strengths of different modeling approaches and to ensure that the designs satisfy all the desired structural, biophysical, enzymatic, and/or therapeutical specifications. One way this can be achieved is through post hoc filtering, whereby a generative model is used to propose designs, which are then scored using one or more different models to screen for designs with desired characteristics. The success of this strategy critically depends on the degree of overlap between the region of the design space explored by the generative model, and the regions of the design space favored by the filters. A low degree of overlap leads to a high rejection rate and likely sub-optimal design candidates.

A common cause of low overlap is that the filters conflict with each other or the generative model, and thus cannot be simultaneously optimized. This type of scenario is prevalent in protein design; for example, foldswitching proteins (also known as metamorphic proteins [45]), or more generally, proteins that undergo conformational changes necessary for their function, may require, at a given position, different sidechain hydrophobicities in the different conformational states, resulting in the impossibility of simultaneous optimization of the folding free energies of all relevant states. To fully account for such tradeoff conditions requires an explicit approximation of the Pareto front in the objective space. A key property of the Pareto front is that the solutions on the Pareto front dominate those outside of the Pareto front; more precisely, for any feasible solution $B$ outside of the Pareto front, one can find a solution $A$ on the Pareto front that is no worse than $B$ on all objectives and improves $B$ on at least one objective. In other words, the Pareto front represents the optimal tradeoff conditions among the objective functions, and thus should be considered the optimization target when multiple models and objective functions are needed to specify the design problem. Although it is straightforward to perform a non-dominated sorting during post hoc filtering, there is in general no guarantee that the non-dominated solutions represent a close approximation or unbiased sampling of the Pareto-optimal solutions, especially when they are produced by a generative model not designed to optimize the objective functions.

In this work, we therefore seek an alternative, integrative framework for sequence design that can more closely guide the generative process and fully account for the Pareto front in the

objective space. Specifically, we argue that evolutionary multiobjective optimization algorithms [46], a class of genetic algorithms designed for multiobjective optimization, provide one such suitable framework that can be adapted to meet these criteria. This is because these algorithms are, by construction, designed to explicitly approximate the Pareto front in a user-specified objective space, and, as we demonstrate in this work, can iteratively guide the sampling process through the construction of biophysically-informed mutation operators. This emphasis on both explicit approximation of the Pareto front and use of informative mutation operators distinguishes this work from previous applications of genetic algorithms to protein design [47–53].

Multistate design [54] provides a natural setting to demonstrate how an evolutionary multiobjective optimization framework might function in practice, because the need to explicitly represent multiple states of a protein in this type of design problems directly maps onto the ability of the optimization algorithm to simultaneously consider multiple objective functions. In this work, we begin with a detailed analysis of the two-state sequence design problem for *E. coli* RfaH, using the established Non-dominated Sorting Genetic Algorithm II (NSGA-II) [55] as the optimization framework (Fig 1). RfaH is a difficult model system for multistate design because it is a foldswitching protein that undergoes extensive conformational changes between an N-terminal domain bound all-α (RfaHα) and a dissociated all-β (RfaHβ) C-terminal domain [56], which involves considerable differences in the secondary and tertiary structure and solvent accessibility of its amino acid residues. We then characterize the performance of evolutionary multiobjective optimization methods for higher-dimensional design problems, a known challenge for this class of algorithms [57], using PapD and calmodulin (CaM) as the model systems. For these multistate design problems, we have chosen three well-established deep-learning-based models, each representative of a different modeling approach: the inverse folding model ProteinMPNN [19,20] (henceforth abbreviated as pMPNN), the protein language model ESM-1v [27], and the structure prediction model AlphaFold2 [14,58]. In particular, we use AlphaFold2-based confidence metrics to compute what is known as the AF2Rank composite score [59], and employ it as a measure of folding propensity, without the need for multiple sequence alignments.

Through our benchmark analysis, we present a series of genetic algorithm setups that progressively integrate more biophysical information into the mutation operator, and examine their ability to recover the native sequence, given a set of wild-type (WT) structures. We find that NSGA-II, in conjunction with the basic random resetting mutation operator, is uncompetitive with a direct application of pMPNN. However, by embedding pMPNN and ESM-1v directly into the mutation operator, whereby ESM-1v is used to rank the designable positions and pMPNN is used to redesign the least nativelike residues, the genetic algorithm is able to better approximate the Pareto front in the objective spaces, which translates into designs with reduced sequence entropy and improvement in native sequence recovery, especially at positions where pMPNN alone fails. Together, these results indicate that evolutionary multiobjective optimization is a suitable framework for integrating multiple models directly into the protein sequence design process, and there is considerable value in constructing informative mutation operators to guide and accelerate the exploration of protein sequence space.

## Results

### The random resetting mutation operator results in slow convergence

In this work, we seek to assess whether multiobjective evolutionary optimization algorithms constitute a suitable framework for integrating multiple models and sources of information into protein sequence design, using the two-state design of the foldswitching protein RfaH as an in-depth test case, and PapD and CaM as examples of higher-dimensional sequence design problems (Fig 1B). In general, such a genetic algorithm has three components (Fig 1A): (i) a

A.

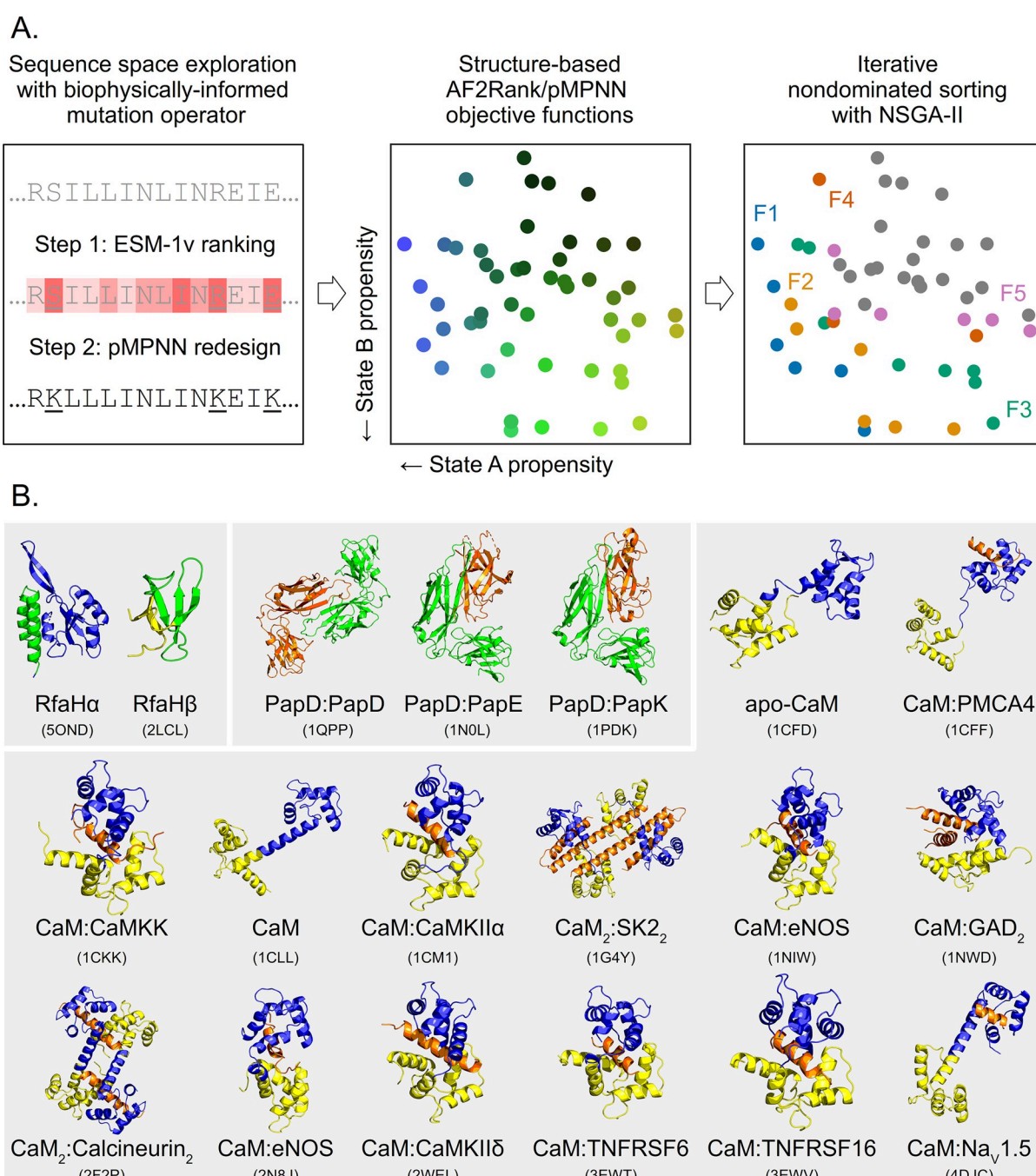

**Fig 1. Evolutionary multiobjective optimization provides a suitable framework for multistate design.** A. In this work, we examine how machine learning models such as pMPNN, AlphaFold2/AF2Rank, and ESM-1v may be integrated directly into protein sequence design through a multiobjective optimization method known as Non-dominated Sorting Genetic Algorithm II (NSGA-II). Left: first, new design candidates are proposed through a mutation operator; here, this operator is composed of ESM-1v, which is used to rank residue positions, and ProteinMPNN (pMPNN), which is used to redesign the least nativelike positions. Middle: the design candidates are then scored using objective functions derived from AlphaFold2 and pMPNN confidence metrics. Right: lastly, the scored candidates are sorted into successive pareto fronts (here numbered F1 to F5), and the candidates from the best fronts are selected by NSGA-II for the next round of design. See Methods and S1 Fig for additional details. B. To demonstrate the effectiveness of this framework, we perform an in-depth analysis of the two-state sequence design problem for RfaH, a small foldswitching protein whose C-terminal domain can interconvert between an all-α RfaHα state and an all-β RfaHβ state. We then examine the ability of the proposed framework to tackle higher-dimensional design problems, by redesigning the multi-specific binding interface of PapD (three states) and the various binding modes of calmodulin (14 states). For RfaH, the designable positions are highlighted in green; note that the N-terminal domain is not shown in the cartoon representation of the RfaHβ state. For PapD, the binding partner is shown in orange, and PapD is shown in green. For CaM, the N-terminal half of the protein (residues 1–74) is

shown in yellow, the C-terminal half of the protein (residues 75–148) is shown in blue, and the binding partners, when present, are shown in orange. For each structure, its PDB ID is listed in parenthesis. The CaM binding partner name abbreviations are explained on Table 1.

method for proposing new candidates, typically through the use of (binary) tournament selection, crossover operators, and mutation operators, (ii) a set of objective functions for scoring the proposed candidates, and (iii) an algorithm for forming a new population from the existing and proposed candidates. For the first two components, we will investigate how pMPNN, AF2Rank, and ESM-1v can be incorporated into sequence design through the mutation operators and objective functions (for the crossover operator, we find that the number of crossover points has a minimal impact on the design outcome (S3 Fig); therefore, for all following simulations, we set the number of crossover points to two and focus on the mutation operator instead). For the third component, we employ NSGA-II [55], an established genetic algorithm for multiobjective optimization (see Methods for more details).

In the following benchmark analysis, we will employ pMPNN both as an objective function and as a generative model in the context of both single-state and multistate design. For the sake of clarity, we will refer to pMPNN as pMPNN-SD (single-state design) when it is used to generate sequences conditioned on the structure of a single state, and pMPNN-AD (average decoding) when it is used to perform multistate design by averaging logits over multiple states during sequence decoding; when pMPNN is used as an objective function, we will refer to its log likelihood score for the full sequence, conditioned on the structure of a single state, as the pMPNN-SD log likelihood score. In contrast, sequences designed using NSGA-II will always be referred to with the label GA (genetic algorithm), followed by a description of the simulation setup in square brackets.

We begin with the simplest baseline setup using the random resetting operator: starting with a population of fully randomized RfaH C-terminal domain sequences (residues 119 to 154), with a probability controlled by the mutation rate parameter, each designable position is selected for redesign, and the redesign is done with uniform sampling over the set of standard (20 naturally occurring) amino acids. The redesigned sequences are then scored against the structures of the RfaHα and RfaHβ state, with either the pMPNN-SD log likelihood or the AF2Rank composite score (see Methods). This setup is closely related to protein design via hallucination [60,61], and, in a sense, inverts the learned sequence-to-structure mapping of AlphaFold2; similar methods have been explored in the literature, typically with unsatisfactory results [20,21,43]. Consistent with the literature, we find that this baseline setup is slow to converge by all considered metrics (Fig 2, first two columns), fails to reach parity with pMPNN-AD (Fig 2, gray dotted lines), and the resulting sequence profiles exhibit a high degree of sequence entropy, especially for those designed using the AF2Rank objective functions (S4 and S5 Figs), indicating that these simulations have not converged in the sequence space. These results suggest that a naive application of a genetic algorithm is likely to be outperformed by a single-pass inverse folding model, and AlphaFold2 is, on its own, a poor mutational effect predictor [59,62]. Indeed, we find that the the sequences designed using the AF2Rank objective functions, especially those at low mutation rates, have AF2Rank composite scores comparable to those of the WT sequence (S6 Fig), despite the fact that native sequence recovery from these simulations is close to zero (Fig 2, second column).

## A pMPNN-based mutation operator improves autoregressive sequence decoding

Next, we investigate whether the poor performance of the baseline setup can be rescued with a more informative mutation operator. From this point on, we will refer to each GA setup with

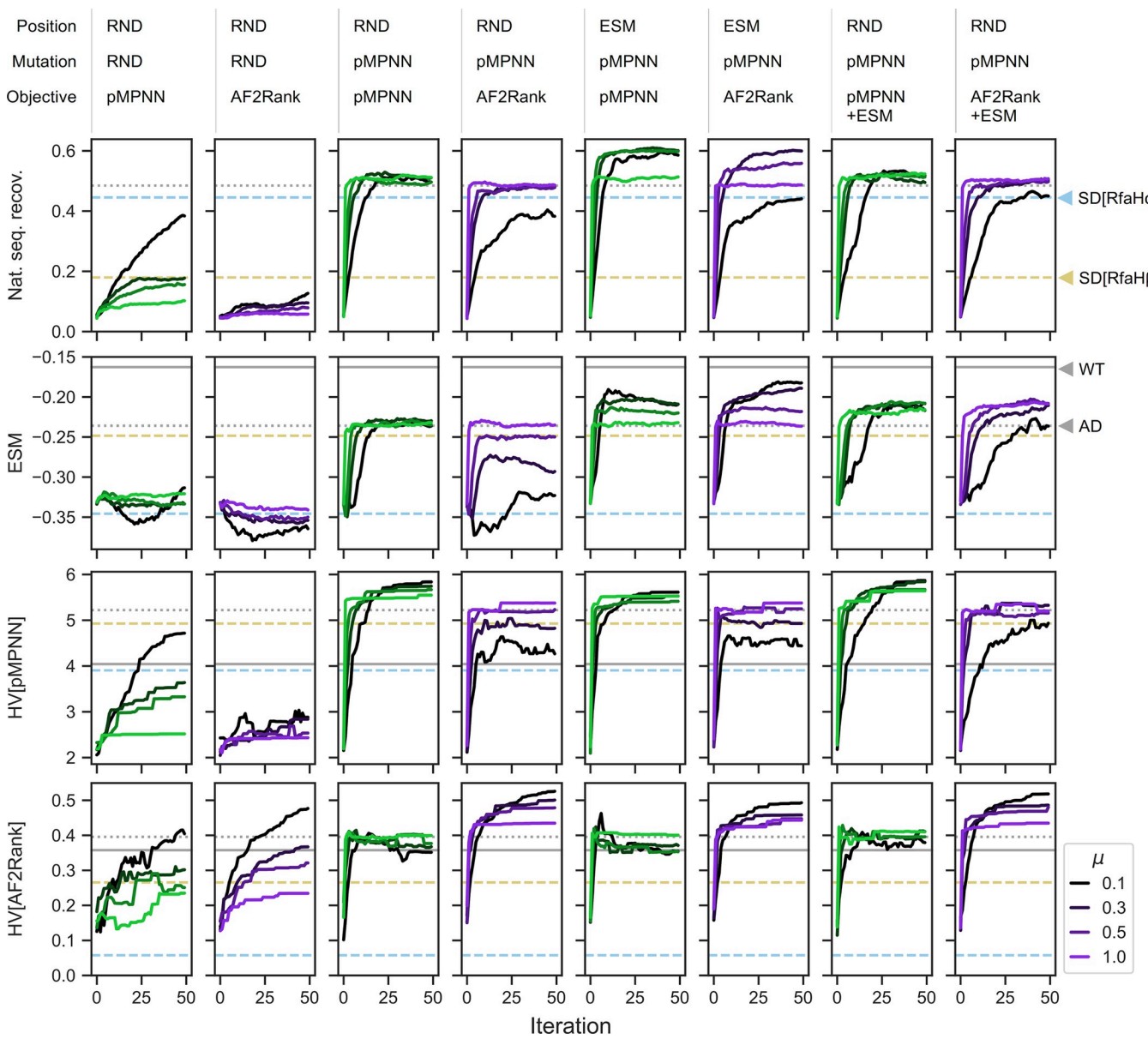

**Fig 2. Integration of multiple models through evolutionary multiobjective optimization improves RfaH sequence design outcomes.** Each column represents one genetic algorithm setup, indicated by the legend above each column that describes how, at each iteration, the subset of designable positions is selected, how the mutations are proposed, and which objective functions are used to score the designs; here, RND stands for random, pMPNN stands for ProteinMPNN, and ESM refers to ESM-1v. Each row represents a different quality metric: "nat. seq. recov." refers to the population average fractional identity to the WT sequence, "ESM" refers to the ESM-1v log likelihood score, averaged over both the population and sequence positions, "HV" refers to hypervolume in the pMPNN-SD objective space (HV[pMPNN]), and hypervolume in the AF2Rank composite score objective space (HV[AF2Rank]); see Methods for more details. Each panel represents the progression of a genetic algorithm simulation over 50 iterations and at four different mutation rates ($\mu$). Within each panel, the horizontal lines represent the quality metric values calculated for the WT RfaH sequence (solid gray), the population average of the pMPNN-AD design sequences (dotted gray), the population average of the RfaHα pMPNN-SD design sequences (dashed blue), and the population average of the RfaHβ pMPNN-SD design sequences (dashed yellow). Simulations results at additional mutation rates are shown in S2 Fig.

three descriptors: how the subset of designable positions is selected, how the mutations are proposed, and how the designed candidates are scored; for example, the baseline setup with the random resetting operator and the AF2Rank objective functions can be denoted this way as GA[RND,RND,AF2Rank], where RND stands for random.

While the random resetting operator allows unbiased exploration of the sequence space, it is inefficient, given the combinatorial complexity of the protein sequence space. On the other hand, inverse folding models such as pMPNN have likely learned a lower-dimensional manifold in the sequence space, conditioned on the backbone conformation, that allows for efficient and accurate decoding of native-like sequences. Motivated by this observation, we modify the random resetting operator, so that the selected subset of designable positions are passed onto pMPNN-AD for redesign. We note here that this setup is similar to a recent method [53] that incorporated pMPNN into a genetic algorithm, although in [53], pMPNN is used in addition to, rather than in place of, the random resetting operator, and each application of pMPNN modifies all designable positions, rather than a subset thereof. With this modification, for at least some choices of mutation rate, the genetic algorithm is able to reach parity with pMPNN-AD in terms of native sequence recovery and the average ESM-1v log likelihood score, and outperforms pMPNN-AD in terms of hypervolume [63], a metric that assesses the optimality and diversity of candidate sequences (see Methods) (Fig 2, third and fourth columns; compare with the gray dotted lines).

The GA[RND,pMPNN,pMPNN] setup provides a useful case study in the potential advantages of genetic algorithms. This specific setup injects no additional biophysical information into the sequence design process, yet the resulting Pareto front shows clear separation in the pMPNN-SD objective space from the population generated with pMPNN-AD (S7 Fig). This result demonstrates the potential advantage of the evolutionary multiobjective optimization framework over the post hoc filtering approach to sequence design: although it is possible to approximate the Pareto front in the objective space by a post hoc non-dominated sorting from the pMPNN-AD population, the resulting approximate Pareto front will still be dominated by the design population generated by the genetic algorithm.

What could explain the performance difference between pMPNN-AD and GA[RND, pMPNN,pMPNN]? This difference is partly explained by the fact that a genetic algorithm is a numerical optimization technique, as compared to the autoregressive Monte Carlo sampling steps of pMPNN; as such, suboptimal residue types that are selected against in a genetic algorithm may be retained, even if only with low probability, by pMPNN-AD alone. More importantly, this difference rests on a basic limitation of autoregressive decoding: the first positions to be decoded tend to have the highest level of uncertainty, because more of the sequence context remains masked initially. Here, the genetic algorithm setup alleviates this problem in two ways: by reducing the number of masked positions in each pass, and by revisiting the same positions over the iterations, presumably with better sequence context each time. More precisely, with the GA[RND,pMPNN,pMPNN] setup, the number of redesigned positions with each application of the mutation operator and the number of times a position is redesigned both follow binomial distributions. This explanation is consistent with the observation that the gap between the GA[RND,pMPNN,pMPNN] sequences and the pMPNN-AD sequences narrows as the mutation rate increases (S7 Fig), because a higher mutation rate increases the number of masked positions in each pass, and it weakens the Markovian dependence between successive iterations.

## ESM-1v accelerates sequence space exploration

Lastly, we examine the effect of incorporating ESM-1v into the genetic algorithm. Given that ESM-1v is a good variant/mutational effect predictor [27], we propose two approaches to incorporating it into the design process. First, similar to a method described in [64] that prioritizes sequence design over low AlphaFold2 confidence regions, we use the ESM-1v marginal probabilities to rank and select the least nativelike positions for redesign (Fig 2, fifth and sixth

columns); second, we directly incorporate the per-sequence average ESM-1v log likelihood score as a third objective function (Fig 2, last two columns). We find that the first approach (GA[ESM,pMPNN,pMPNN/AF2Rank]) produces the overall best performance, while the second approach (GA[RND,pMPNN,pMPNN/AF2Rank+ESM]) shows worse performance in terms of both native sequence recovery and average ESM-1v log likelihood score.

What could account for the performance differences between these two approaches? The difference in native sequence recovery can be rationalized by examining the pMPNN-SD design results (Fig 2): on average, the RfaHα single-state redesigned sequences are more similar to the pMPNN-AD sequences in terms of native sequence recovery, but the RfaHβ single-state redesigned sequences are more similar to the pMPNN-AD sequences in terms of the ESM-1v log likelihood score. As such, setting the ESM-1v score as an objective function biases sampling towards RfaHβ-like sequences with lower identity to the WT (S8 Fig). In contrast, in the GA[ESM,pMPNN,pMPNN/AF2Rank] setups, because ESM-1v is not used to generate mutations, the effect of its preference for the RfaHβ state is much weaker on native sequence recovery. In theory, the bias could have slowed down, or even stalled, convergence, by directing computation towards positions that are already nativelike, but not RfaHβ-like; however, comparison of the rate of convergence across the setups do not support this hypothesis (Fig 2).

The difference in the average ESM-1v log likelihood scores can be rationalized in terms of the structure of the simulation setups. In the second approach, optimization of ESM-1v as a third objective function is constrained by tradeoffs with the need to simultaneously optimize the pMPNN-SD log likelihood scores or AF2Rank composite scores, which do not appear to be correlated with ESM-1v scores (S9 Fig). In contrast, in the first approach with the GA[ESM, pMPNN,pMPNN/AF2Rank] setups, because ESM-1v is used to set the design positions, the simulation prioritizes refinement of the lowest-ranked positions, which necessarily leads to an improvement in the sequence-averaged ESM-1v scores. As such, ESM-1v is invoked upstream of non-dominated sorting in the objective space, and the indirect optimization of its scores is not subject to the same tradeoff conditions in the second approach.

Interestingly, the effect of ESM-1v in the GA[ESM,pMPNN,AF2Rank] setup can be interpreted in the framework of Bayesian inference:

$$p(\text{seq}|\text{str}) \propto p(\text{str}|\text{seq})p(\text{seq})$$

where the posterior distribution $p(\text{seq}|\text{str})$ is sampled by pMPNN-AD during sequence decoding, the likelihood function $p(\text{str}|\text{seq})$ is approximated by the AF2Rank composite score, and the prior $p(\text{seq})$ is partially set by the mutation operator. The random resetting operator is equivalent to a flat prior, and gives every designable position equal priority for redesign. The use of ESM-1v to rank and select positions for redesign can be interpreted as an informative prior that accelerates traversal in the sequence space.

## RfaH multistate design shows sequence tradeoff patterns based on solvent accessibility

A key feature of an evolutionary multiobjective optimization algorithm is that the solutions approximate the Pareto front in the objective space. To the extent that objective functions such as the pMPNN-SD log likelihood score and the AF2Rank composite score correlate with desirable biophysical properties such as stability, this Pareto optimality condition necessarily needs to manifest in the sequence space as mutations that improve such properties for either or both states of RfaH. Therefore, in this and the following sections, we investigate in detail the differences in the sequence profiles generated by pMPNN-SD, pMPNN-AD, and the genetic algorithms, and the potential biophysical consequences of such differences. Among the genetic

algorithm setups we have examined so far, GA[ESM,pMPNN,pMPNN/AF2Rank] give the best overall performance, especially at the intermediate mutation rate 0.3; as such, we will focus on the sequences generated in the last iteration of these two sets of simulations in this analysis. For the sake of brevity, we will simply refer to these two setups and their last iteration sequences as GA[pMPNN] and GA[AF2Rank].

To understand the distribution of designed sequences in the sequence space, we first seek a low dimensional representation of the RfaH C-terminal domain sequence space, using BLO-SUM62 as a similarity metric and Laplacian eigenmaps [65] as the dimensionality reduction technique (Fig 3A, first panel). This analysis reveals three distinct sequence clusters: one occupied by the RfaHα pMPNN-SD design results (light blue), one occupied by the RfaHβ pMPNN-SD design results (yellow), and one occupied by the WT RfaH sequence (black), the pMPNN-AD design results (gray), and the genetic algorithm design results (green and purple). This clustering pattern is largely reproduced in the pMPNN-SD and AF2Rank objective space (Fig 3A, second and third panels), although there is a greater spread of sequences in the AF2Rank objective space, which appears to indicate a somewhat greater sensitivity of the AF2Rank composite score to the differential sequence preferences of the two RfaH states (S10 Fig). Nevertheless, the fact that all three embeddings generate similar clustering patterns suggest that the multistate design methods examined so far have all succeeded, to various extent, in recapitulating the WT RfaH sequence profile, which is distinct from hypothetical non-foldswitching single-state RfaH sequence profiles.

Next, we investigate the sequence composition underlying the apparent separation between the single-state and multistate design results. This analysis reveals two distinct but equally prevalent patterns in the sequence profiles generated through pMPNN-SD and pMPNN-AD (Fig 3B): conflicting residue positions (first two panels), where the most frequently recommended residue types from one of the two states dominates the multistate sequence profile, and non-conflicting residue positions (third panel), where the multistate sequence profile consists of a union of residue types assigned to both states. In addition, for the 17 positions where the two states have conflicting residue preferences, 13 are dominated by the RfaHα state, and these 13 positions constitute the majority of buried core residues in this state (yellow shading, middle row), while the majority of the RfaHβ core residues (yellow shading, bottom row) are located at non-conflicting positions. Interestingly, of the RfaHα core positions that dominate the multistate sequence preference, all but E149 are located at the N- and C-terminal domain interface. These observations are consistent with the fact that the RfaHα state is highly stabilized when in contact with the N-terminal domain, while the RfaHβ state is the preferred, albeit only marginally stable, state as the two domains dissociate [66]. Overall, this pattern of sequence preference explains the clustering patterns between the single-state and multistate design sequences, and why the multistate design sequences appear to be closer to the RfaHα single-state design sequences in the low-dimensional representation of the RfaH sequence space. Lastly, we note here that, although it is theoretically possible for the multistate sequence profile to be distinct from both single-state designs (e.g., a foldswitching protein may contain a polar residue at a position where its two constituent conformational states may respectively prefer charged and hydrophobic residues), this third scenario is not observed with the current model system.

## Genetic algorithms can reduce bias and variance in native sequence recovery

Having examined closely the difference among the sequences generated using pMPNN-AD and pMPNN-SD, we now turn to the difference among the sequences generated using

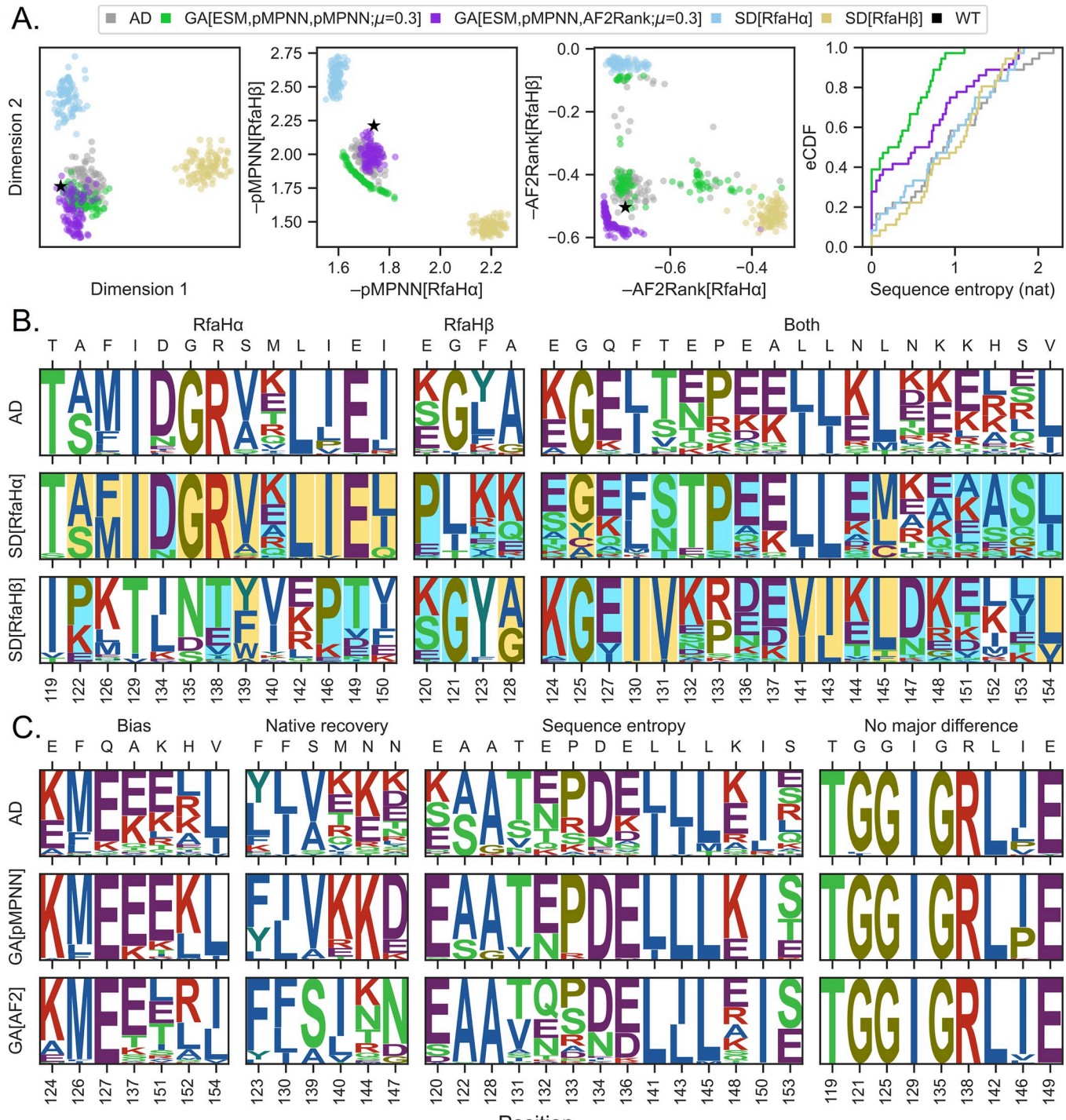

**Fig 3. Integration of multiple models into RfaH sequence design leads to reduction of sequence bias and variance.** A. From left to right: distribution of pMPNN-AD (gray), pMPNN-SD (blue for the RfaHα state and yellow for the RfaHβ state), GA[ESM,pMPNN,pMPNN;μ = 0.3] (green; later abbreviated as GA [pMPNN]), GA[ESM,pMPNN,AF2Rank;μ = 0.3] (purple; later abbreviated as GA[AF2Rank]), and the WT (black) sequences in a two-dimensional embedding generated with the Laplacian eigenmaps algorithm [65], the pMPNN-SD log likelihood objective space, and the AF2Rank composite score objective space. The fourth panel on the right shows the empirical cumulative distribution functions (eCDF) for the per-position sequence entropy (base *e*) of these populations of designed sequences. All GA sequences in this figure refer to the final iteration sequence populations. B. Logo plots for sequences generated using pMPNN-AD (top), pMPNN-SD for the RfaHα state (middle), and pMPNN-SD for the RfaHβ state (bottom). The residue positions are organized into three blocks, depending on whether the sequence profiles from the RfaHα (left), RfaHβ (middle), or both states (right) dominate the pMPNN-AD sequence profiles. For the RfaHα and RfaHβ pMPNN-SD design sequence logo plots, the residue positions are shaded according to their relative solvent accessibility in the

corresponding state; blue shading indicates a relative solvent accessibility > 50%, while yellow shading indicates < 20%. The WT residue type at each position is indicated on the secondary x-axis. C. Logo plots for sequences generated using pMPNN-AD (top), GA[pMPNN] (middle), and GA[AF2Rank] (bottom). The residue positions are organized into four blocks: "bias", "native recovery", "sequence entropy", and "no major difference"; see the main text for more details on this classification.

pMPNN-AD, GA[pMPNN], and GA[AF2Rank]. This leads us to divide the set of designable positions into four groups (Fig 3C): "bias", where the native residue is recovered at a very low level, if at all, with all three setups; "native recovery", where GA[pMPNN] and/or GA[AF2Rank] are able to recover the native or native-like residue at significantly higher fraction than pMPNN-AD; "sequence entropy", where all three methods are able to frequently recover the native residue, but GA[pMPNN] and/or GA[AF2Rank] show lower level of sequence entropy; and "no major difference", where the three methods have identical or near-identical performance.

Setting aside the "no major difference" category, most of the designable positions differ in terms of sequence entropy in the sequence populations generated using pMPNN-AD, GA[pMPNN], and GA[AF2Rank] (Fig 3C, third column). In particular (Fig 3A, fourth panel), the sequence population generated by GA[pMPNN] (green) tends to have the lowest level of sequence entropy, followed by that generated using GA[AF2Rank] (purple), while the sequences generated using pMPNN-AD (gray) and -SD (light blue and yellow) tend to have the highest level of sequence entropy; this pattern holds for most of the genetic algorithm set-ups examined in this work (S5 Fig). This reduction in sequence entropy oftentimes concentrates probability on the WT residue types, thus partially explaining why GA[pMPNN] and GA[AF2Rank] have much better native sequence recovery than pMPNN-AD (Fig 2, fifth and sixth columns). Given this observation, we ask whether it is possible to achieve similar sequence entropy reduction by performing a post hoc non-dominated sorting of the pMPNN-AD sequences in the objective spaces; this filtering step results in a significant loss of the sequence population, yet the resulting sequence entropy profiles are no better than that of a random subsampling of the sequence population (S11 Fig), which indicates that the post hoc approximate Pareto front does not confer the same benefit as that generated using true multi-objective optimization.

The "bias" and "native recovery" categories (Fig 3C, first two columns) suggest systematic differences between the RfaH WT sequence and the structure-sequence mapping learned by pMPNN. In particular, at residues Q127, A137, M140, N144, N147, and H152, which are positions that are at least somewhat solvent exposed in one of the two states, pMPNN tends to substitute charged residues for the polar and hydrophobic WT residues, which results in increased surface charges and new surface salt bridge interaction networks (S12 Fig). This is a known bias of pMPNN [20], and explains why the WT sequence is scored somewhat unfavorably by pMPNN-SD (Fig 3A, second panel) compared to the redesigned ones.

At F130, S139, M140, N144, and N147, the WT residues (or, in the case of M140, physico-chemically similar residues) are recovered only through GA[AF2Rank]. Among these residues, the effect of charged substitutions such as K and E at M140 and N144 is ambiguous, because such substitutions tend to have lower β-strand propensity, even though an increase in surface charges and salt bridge interactions may improve stability and facilitate strand pairing during folding [67]. On the other hand, F130 and S139 likely play an important role in shaping the free energy landscape of RfaH, in a way that likely cannot be replaced by the mutations suggested by AD and GA[pMPNN]. First, in the RfaHα state, F130 is located between the N- and C-terminal domains and mediate their interface (S13A Fig), and protease accessibility assays have demonstrated that F130V, a substitution to a smaller hydrophobic residue, weakens such

interface interaction [68]. Second, S139 is located at α2 in the RfaHα state and β3 in the RfaHβ state. In the RfaHα state, S139 introduces a buried unsatisfied hydrogen bond donor and acceptor into the N- and C-terminal interface (S13C Fig), while in the RfaHβ state its sidechain forms a hydrogen bond interaction with that of N156 near the core of the domain (S13B Fig); as such, a mutation to V or A is expected to further stabilize the RfaHα state at the expense of the RfaHβ state. Therefore, at residue position 139, GA[pMPNN] designs appear to favor the RfaHα state, while the GA[AF2Rank] designs tend to maintain the WT serine residue. However, in the WT, the N-terminal portion of α2 already exhibits high local stability [69] and transient helical content in the foldswitching transition state ensemble [70,66], likely helped by its high leucine content (L141, L142, L143, and L145) [71], while the C-terminal all-β state is only marginally stable, even though it is the energetically preferred state when the C-terminal domain is dissociated from the N-terminal domain [66]. Therefore, by maintaining WT residues, the GA[AF2Rank] designs are more likely to retain the foldswitching phenotype of the WT sequence compared to the GA[pMPNN] designs. These results therefore highlight how integrating multiple models into the sequence design process can produce sequences that may capture key energetic effects better than what each model can produce individually.

## pMPNN hyperparameter tuning leads to bias-variance tradeoff

In the previous section, we demonstrated how the Pareto optimality condition in the objective space can manifest in the sequence space both in terms of bias reduction (i.e., concentration of probability on the WT residue types) and variance/entropy reduction (i.e., concentration of probability on a smaller set of residue types). This result is achieved without any finetuning of pMPNN hyperparameters, such as temperature or state weights; in this section, we analyze whether modifications to these hyperparameters might yield similar improvements without invoking the evolutionary multiobjective optimization framework.

First, we examine whether it is possible to reduce pMPNN-AD sequence entropy by lowering the sampling temperature. To answer this question, we perform additional sequence design with pMPNN-AD at the temperature of 0.1, and compare the results with existing simulations performed at temperature 0.3. As expected, reducing the sampling temperature reduces sequence entropy (S14B Fig, fourth panel), which leads to an improvement in native sequence recovery (S14A Fig). This is manifested in the pMPNN-SD objective function space as a partial overlap between the higher temperature Pareto front and the lower temperature pMPNN-AD population (S14B Fig, second panel). However, in the AF2Rank objective space, lowering the temperature moves the pMPNN-AD population further away from the GA[AF2Rank] Pareto front (S14B Fig, third panel). Consistent with this observation, at this lower temperature, pMPNN-AD no longer recovers the native sequence at positions Q127, S139, and N144, and recovers the native sequence with a much lower probability at positions F123 and F126 (S14C Fig). As such, while lowering the temperature reduces variance, it also increases the bias in the redesigned pMPNN-AD sequences.

Next, we examine the effect of state weights in native sequence recovery. In pMPNN-AD, the model takes a weighted average of the state-dependent logits during autoregressive sequence decoding, with the default weight being 1.0 for each state (before normalizing the sum of all state weights to 1.0). As we saw with Fig 3B, the pMPNN-AD sequence profiles are the results of balancing the oftentimes conflicting sequence preferences of each constituent state. Therefore, one plausible explanation for why pMPNN-AD and GA[pMPNN] sometimes fail to recover the native sequence is that there is a mismatch between the state weights and the actual local stability difference between the two states of WT RfaH. To assess this hypothesis and whether it can be corrected by rebalancing the state weights, we extract the raw logits

(conditioned on the rest of the WT sequence) for positions in the "native recovery" group in Fig 3C, and use the logits to compute how the pMPNN-AD sequence profiles at these positions change as a function of the state weights (S15 Fig). We find that for F130, native recovery could have been significantly improved by a stronger weight for the RfaHα state, although this does come with increased sequence entropy. For S139, M140, N144, and N147, however, varying the state weights would not have meaningfully improved native sequence recovery, because the native residue is never among the preferred residue types under any state weight choices; the fact that GA[AF2Rank] succeeds in recovering the native sequence at many of these positions highlights the ability of the genetic algorithm to pick out and amplify small signals, when equipped with the appropriate objective functions. Taken together, it appears that adjusting state weights is not a useful strategy for correcting biases in native sequence recovery; even in cases such as F130 where the state weight appears to have been misspecified, it is unclear how, a priori, one should set the per-position state weights, especially considering the fact that these weights do not have a straightforward biophysical interpretation.

In conclusion, if sequence design is viewed as a statistical inference problem, the effect of tuning the pMPNN state weights and sampling temperature is subject to the bias-variance tradeoff, where hyperparameter choices that reduce bias come at the cost of increased variance, and vice versa. This is to be contrasted with the results from the previous section, where we showed that GA[pMPNN] and GA[AF2Rank] can reduce bias and/or variance in the population of redesigned sequences, through the integration and Pareto optimization of multiple sources of information on the RfaH sequence-structure relation.

## Genetic algorithms improve sequence similarity to RfaH-like foldswitching proteins

Even though native sequence recovery is a standard metric for assessing the performance of sequence design methods, the metric may have difficulty capturing the sequence patterns underlying the foldswitching capacity, or lack thereof, of the redesigned RfaH sequences. This assessment is based on the estimation that the average pairwise sequence identity of foldswitching sequences related to the NusG superfamily, to which RfaH belongs, is only around 20% [72]. Given this observation, we seek an additional metric to assess how well the RfaH sequences redesigned by the multistate methods might retain the RfaH foldswitching phenotype. To do so, we compare the redesigned sequences to a computationally annotated and experimentally validated database of NusG-like sequences [72], which are predicted to be either foldswitchers or non-foldswitchers (see Methods and Fig 4). We find that sequences

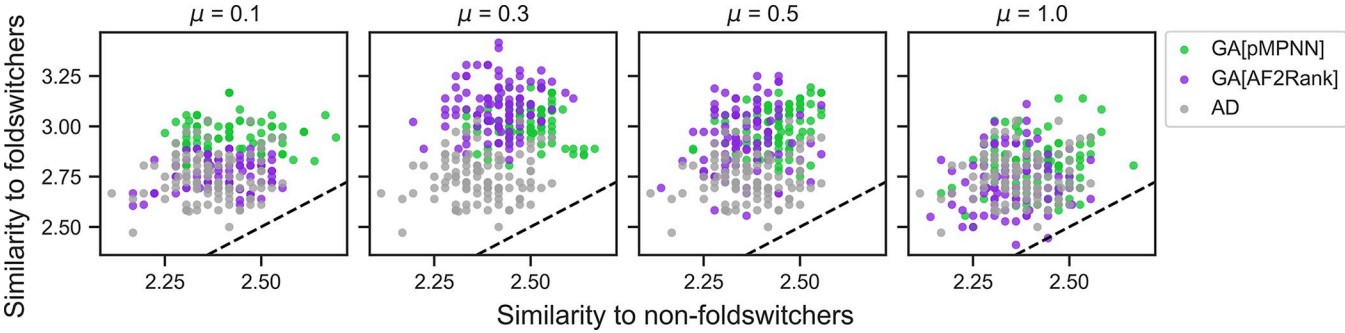

**Fig 4. Genetic algorithm RfaH designs exhibit greater sequence similarity to NusG-like foldswitching proteins than non-foldswitching proteins.** The four panels show the distribution of sequence similarity measures for the pMPNN-AD (gray), GA[pMPNN] (green), and GA[AF2Rank] (purple) sequences to NusG-like foldswitching and non-foldswitching sequences. Each panel corresponds to a different mutation rate for the GA simulations, indicated in the panel title. The dashed black lines represent the $y = x$ lines. See Methods for more details on the similarity measure and the NusG-like sequence database.

designed by GA[pMPNN] and GA[AF2Rank] tend to be more similar to the foldswitcher sequences than those designed using pMPNN-AD, while the similarity to the non-foldswitchers is largely unaffected. However, the extent to which the genetic algorithm outperforms pMPNN-AD depends on the mutation rate, and, again, the mutation rate 0.3 appears to produce the best sequences. These results are consistent with our analysis of the GA[pMPNN] and GA[AF2Rank] sequences so far, and confirm that these methods have the potential to outperform pMPNN-AD for multistate protein sequence design.

## Genetic algorithms can be applied to higher-dimensional design problems

In this work, we have so far focused on the protein RfaH as an in-depth case study of how evolutionary multiobjective optimization algorithms can be used to integrate multiple models into the protein sequence design process. RfaH is a difficult design target because of its extensive conformational changes during foldswitching. On the other hand, by only explicitly modeling the two end states of its foldswitching transition, we have treated the RfaH sequence design problem in a two-dimensional objective space. Such low-dimensional optimization problems are well-studied in the field of evolutionary multiobjective optimization, but applications of evolutionary multiobjective optimization techniques to higher dimensional problems may be difficult because the scaling of objective space volume with dimensionality may potentially require exponentially larger population sizes to adequately represent the Pareto front.

In this section, we benchmark the genetic algorithm setups examined so far against two additional model systems of higher dimensions: the *E. coli* P pilus chaperone protein PapD [73] and the vertebrate calcium-binding protein CaM [74], following previous works on multistate design [75,76]. For PapD, we redesign its multi-specific interface between PapD-PapE, PapD-PapK, and PapD-PapD, which results in a three-dimensional objective space; for CaM, we model a wide variety of functional states that involve changes in its homo-oligomerization states, central linker conformation, binding partners, and calcium-binding state, which results in a 14-dimensional objective space, and redesign almost all but the $Ca^{2+}$ binding residues (see Methods for more details). Given the observed performance of GA[ESM,pMPNN,pMPNN] and GA[ESM,pMPNN,AF2Rank] at mutation rates near 0.3 for RfaH (Fig 2, fifth and sixth columns), we restrict our analysis to these two setups only, while varying the mutation rates between 0.1, 0.3, and 0.5 and leaving the other genetic algorithm hyperparameters unchanged from the RfaH benchmarks. Continuing with the abbreviations used in the analysis of RfaH, we will refer to these two setups at the mutation rate 0.3, and their last iteration sequences, as GA[pMPNN] and GA[AF2Rank].

For PapD, we find that both GA[pMPNN] and GA[AF2Rank] outperform pMPNN-AD in terms of native sequence recovery and the average ESM-1v log likelihood score (Fig 5A, first two columns). However, unlike RfaH, where the two setups have similar performances according to these two metrics, GA[pMPNN] exhibits better performance for PapD than GA[AF2Rank]; this difference appears to be primarily due to the higher sequence entropy of the GA[AF2Rank] sequence population, which in this case is comparable to that of pMPNN-AD (Fig 5B, right panel), rather than biases in native sequence recovery unique to GA[AF2Rank] (S16A Fig). Interesting, both GA[pMPNN] and GA[AF2Rank] appear to strongly favor methionine for residue 1 in PapD (S16A Fig), even though the native residue at this position is A, due to the cleavage of the N-terminal signal peptide [73]. The preference for methionine as the first residue in polypeptide chains has also been observed for ESM-IF1 [77]; this is likely a common feature of inverse folding models, which is in this case amplified by the genetic algorithm.

In terms of hypervolume, however, pMPNN-AD appears to outperform the genetic algorithm setups in the AF2Rank objective space (Fig 5A, last two columns). Analysis of the

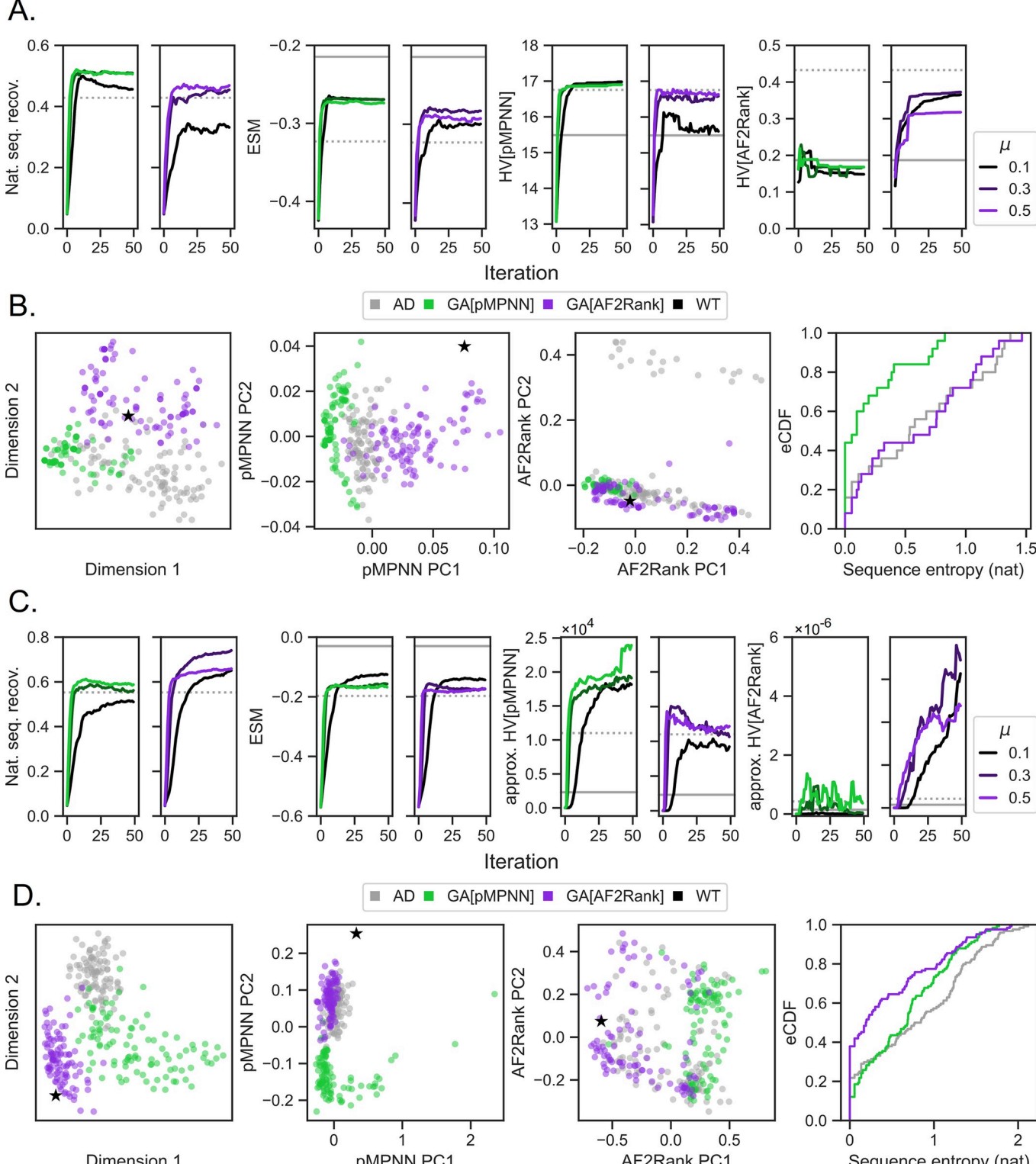

**Fig 5. Genetic algorithms can be applied to higher-dimensional design problems.** The GA[ESM,pMPNN,pMPNN] and GA[ESM,pMPNN,AF2Rank] setups are applied to two additional model systems: PapD (3 states) and CaM (14 states). A and B show the benchmark results for PapD, and C and D show the benchmark results for CaM. Similar to Fig 2, A and C show simulation progression as measured by four quality metrics (from left to right): native sequence recovery, ESM-1v log likelihood scores, pMPNN-SD log likelihood score hypervolume, and the AF2Rank composite score hypervolume. For each quality metric, the panel with green curves represent the GA[ESM,pMPNN,pMPNN] setups, the panel with purple curves represent the GA[ESM,pMPNN,AF2Rank] setups,

and the horizontal lines within each panel represent the WT values (solid gray) and the pMPNN-AD population averages (dotted gray). Note that the ESM-1v scores are computed over the PapD and CaM sequences only, without the binding partner sequences. For CaM, "approx. HV" indicates that the hypervolume metrics are computed using an approximate algorithm (see Methods). Similar to Fig 3A, B and D show distributions of the redesigned sequences in the sequence and objective spaces and their sequence entropies; as in Fig 3, the GA[ESM,pMPNN,pMPNN;$\mu$ = 0.3] and GA[ESM,pMPNN,AF2Rank;$\mu$ = 0.3] setups are abbreviated as GA[pMPNN] and GA[AF2Rank], respectively. Because of the higher dimensionality of the objective spaces, the middle two panels in B and D show the first two principal components (PC) from a principal component analysis of the pMPNN-SD log likelihood objective space and the AF2Rank composite score objective space.

sequence distributions in the AF2Rank objective space indicates a bimodal distribution of the pMPNN-AD population (Fig 5B, third panel), with a minor pMPNN-AD subpopulation not covered by sequences redesigned with either genetic algorithm setups, which explains the observed difference in hypervolume. Comparison of the subpopulation logo plots suggest that the minor subpopulation is enriched in residues K and R at L107, which are rarely, if ever, found in the GA[pMPNN] and GA[AF2Rank] populations (S16A Fig). Structural modeling of such substitutions at L107 suggest that these residues likely introduce buried charge groups with no readily available salt bridge interactions into the PapD-PapE and PapD-PapK interfaces (S16B Fig, first two panels), while the homodimeric PapD-PapD interface likely tolerates the substitutions, as the basic residues can form solvent-exposed electrostatic interaction with E92 at the dimer interface (S16B Fig, third panel). These results echo the RfaH sequence analysis and again suggest that integrating multiple models through evolutionary multiobjective optimization may result in designs that better capture key energetic effects than individual models alone.

## GA[AF2Rank] recapitulates the CaM charge profile

Next, we examine the sequence design problem of CaM. Given its 14-dimensional objective space, we turn to NSGA-III, an extension of NSGA-II intended for higher-dimensional optimization problems (see Methods), as the genetic algorithm framework.

With NSGA-III, both GA[pMPNN] and GA[AF2Rank] outperforms pMPNN-AD in terms of native sequence recovery, average ESM-1v log likelihood score, as well as hypervolumes (Fig 5C); notably, GA[AF2Rank] produces the best native sequence recovery (Fig 5C, first two columns), which is explained by its lower sequence entropy distribution (Fig 5D, fourth panel) and reduced biases (S17 Fig). Further analysis of the sequence distribution in the objective spaces suggests a clear separation between the sequences generated by GA[pMPNN] and those generated by either GA[AF2Rank] or pMPNN-AD, with the latter closer to the WT sequence in the pMPNN-SD log likelihood score objective space (Fig 5D, second panel). This separation can be explained by differences in the number of charged residues: the GA[pMPNN] sequences are substantially more negatively charged than the WT, the pMPNN-AD sequences tend to be slightly more positive than the WT, while the GA[AF2Rank] sequences are most similar to the WT (S18A Fig).

The WT CaM sequence is highly enriched with negatively charged residues (note that we do not consider the $Ca^{2+}$ binding residues for design in this work; see Methods), and its binding partners typically carry positive charges at CaM binding domains [74]. As such, pMPNN-AD designs that neutralize or invert the charge polarity of the designable portion of the protein are likely deleterious to CaM function; indeed, experimental and computational analysis indicates that CaM exhibits a significant drop in affinity to smMLCKp at pH < 4, at which point the CaM carries slightly positive net charge [78]. What is less clear is whether the even more negatively charged designs of GA[pMPNN] can improve CaM function. In these designs, the additional negative charges are concentrated in the linker region connecting the two calcium-binding domains (S18C Fig), which is involved in the binding interfaces in some

of the CaM states examined in this work. Therefore, we perform additional structural analysis with Rosetta on a representative sequence of the GA[pMPNN] population to assess the effect of the negatively charged substitutions on interface energetics (see Methods). This analysis suggests that the additional negative charges may be beneficial for a subset of the examined CaM binding partners (S18B Fig); this result needs to be interpreted with caution, however, as the highly negatively charged surface environment can lead to a significant upshift of sidechain p$K_a$ that makes the binding energetics less sensitive to individual charge substitutions [78].

## The objective spaces of CaM exhibit low effective dimensionality

Given the success of the examined genetic algorithm setups (especially GA[AF2Rank]) for CaM despite its high dimensionality, we ask whether this performance can partly be explained by the possibility that some of the objective space dimensions may be colinear, thus reducing the effective dimensionality of the objective space below the nominal dimensionality given by the number of objective functions. This hypothesis is motivated by three observations: (i) the pairwise TM-scores for the input CaM structures suggest that many CaM states with a canonical binding mode [74] in a collapsed conformation share highly similar global structures (S19A Fig), (ii) the use of dimensionality reduction techniques in evolutionary multiobjective optimization has shown some success in tackling high-dimensional optimization problems [57], and (iii) theoretical analyses have suggested that functional Pareto optimality conditions may lead to a reduction of phenotypic space into lower-dimensional structures [79].

We examine the effective dimensionality of the CaM sequence design problem in two ways. First, we perform a principal component analysis (PCA) on the GA[pMPNN] sequences and the GA[AF2Rank] sequences in the pMPNN-SD log likelihood score objective space and the AF2Rank composite score objective space, respectively, and look for gaps in the eigenvalue spectra that separate out principal components that explain the largest proportion of variance in the objective space (S19B Fig). We find that, for the GA[pMPNN] sequences, the first principal component alone explains more than 80% of the variances in the pMPNN-SD log likelihood score objective space; for the GA[AF2Rank] sequences, the first two principal components combined explain only about 60% of the observed variance in the AF2Rank composite score objective space. The pMPNN-SD log likelihood score objective space thus appears to possess a more linear, lower-dimensional structure than the AF2Rank composite score objective space. Further analysis of the PCA loading plots suggest that, in the AF2Rank objective space, the first principal component is strongly associated with the CaM$_2$:Calcineurin$_2$ complex (PDB ID: 2F2P) (S19C Fig, right panel), which contains a non-canonical peptide binding site formed by a domain-swapped CaM dimer [80], a unique conformation among the input CaM structures, while the second principal component is aligned with CaM states with a canonical binding mode in a collapsed conformation. The PCA loadings in the pMPNN-SD log likelihood objective space does not appear to allow for similar structural interpretations (S19C Fig, left panel). Regardless, if we define the effective dimensionality of the objective space as the number of principal components with eigenvalues above the largest spectral gap, then the CaM objective space examined in this work appears to be only one- or two-dimensional, depending on the metric used to define the space.

Next, we examine the dimensionality of the per-position objective space. To do so, we extract the raw pMPNN logits for the designable positions in CaM, using the same procedure as for RfaH, and compute the Pearson correlation coefficient for each pair of logit vectors at each designable position. The distribution of these correlation coefficients is strongly peaked at one but with a long, continuous tail (S20A Fig). As such, even though many logit vectors are highly correlated, there does not appear to be an obvious common cutoff value across the

design positions for clustering the logit vectors into a smaller number of orthogonal components. Instead, we perform a single-linkage hierarchical clustering of the logit vectors at each designable position, and examine how the correlation coefficient threshold affects the distribution of cluster sizes, which we interpret as the per-position effective dimensionality (S20B Fig). Consistent with the analysis above, we find that, even at a relatively stringent cutoff value of 0.8, the median per-position effective dimensionality is only about three. This number is notably higher than the effective dimensionality of one for the pMPNN-SD log likelihood score objective space, suggesting that the coupling between positions may lead to further reduction in effective dimensionality. The per-position effective dimensionality does not appear to be correlated with the degree of native sequence recovery at the corresponding positions (S20C Fig).

Taken together, we conclude that, despite the high nominal dimensionality of the CaM objective space, the effective dimensionality of the optimization problem is much smaller, which may have contributed to the high native sequence recovery of the genetic algorithm simulations. To the extent that other multistate protein systems exhibit a similar level of degeneracy, the genetic algorithm framework may therefore be readily applicable to sequence design problems with high-dimensional objective spaces.

## Discussion

In this work, we examined the potential of evolutionary multiobjective optimization as an integrative framework for protein sequence design. This framework was chosen because of its ability to explicitly approximate the Pareto front in a user-specified objective space, and the flexibility it affords to construct informative mutation operators to guide sampling in the sequence space. Using the multistate design problem of the two-state foldswitching protein RfaH as an in-depth case study, as well PapD and CaM as examples of higher-dimensional optimization problems, we showed that this approach led to design candidates with reduced bias and variance in native sequence recovery, without the need for post hoc filtering or pMPNN hyperparameter tuning.

Moving forward, as more models become available that capture additional aspects of protein sequence-structure-function relationships, we anticipate such an evolutionary multiobjective optimization framework to be broadly relevant for and readily adaptable to even more complex design tasks. For example, the framework may be adapted for the design of biologics that simultaneously optimizes stability, affinity, specificity, immunogenicity, and pharmacokinetics, or for the construction of entire signaling or metabolic pathways, where multiple biomolecules are simultaneously designed to optimize functional properties.

Other studies have employed Monte Carlo sampling to invert models such as AlphaFold2 for design [20,21,43,51,60,61,64,81]. However, such an approach can generate "adversarial" sequences with poor biophysical properties that nevertheless optimize the objective functions [21]. Indeed, it is likely that the sequences generated by the GA[RND,RND,AF2Rank] setup in this work, especially those at low mutation rates, should also be considered "adversarial", in that they show minimal native sequence recovery (Fig 2, second column), high sequence entropy approaching the theoretical maximum (S5 Fig), but AF2Rank composite scores comparable to those of the WT sequence (S6 Fig). These observations suggest that, despite the innovation in model architecture and availability of curated sequence and structure datasets, it remains challenging to perfectly align the local maxima of a learned sequence-structure mapping with the true protein fitness landscape. Here we argue that setups such as GA[ESM, pMPNN,pMPNN/AF2Rank] are less likely to produce such artifacts, for two reasons. First, when multiple models feed into the generative process, an adversarial sequence can be propagated only if it is overfitted to by all models. This is less likely if the models rely on different

architectures, loss functions, and training data, and as such their errors are unlikely to be consistently aligned. Second, in a multiobjective optimization setting, the Pareto front often does not contain solutions that simultaneously optimize all objective functions; such solutions are commonly known as either the ideal point or the utopia point, and its absence is indicative of underlying tradeoff conditions in the design space. Taken together, the use of informative mutation operators, in conjunction with the Pareto optimization of the objective functions, may minimize adversarial sequences in a way analogous to the voting ensemble technique [82].

Recently, denoising diffusion probabilistic models [35–44] have emerged as a powerful class of generative models, especially for the purpose of de novo protein backbone generation. Such diffusion models share some similarities to the evolutionary multiobjective optimization framework examined in this work. Both classes of methods transform corrupted samples into biophysically realistic ones through a sequence of intermediate distributions; in a diffusion model, this transformation is achieved through a learned, time-dependent transition kernel, while in this work, this transformation is achieved through a time-independent mutation operator, guided by the Pareto optimality condition. In addition, both classes of methods share an interesting connection with autoregressive models. Some discrete diffusion models can be viewed as a generalization of autoregressive models, when the autoregressive sequence decoding steps are viewed as the reverse of a sequential corruption forward process; however, without the restriction of sequential refinement, such diffusion models have the advantage of a parallel, iterative sampling process [83]. As we demonstrated in the analysis of the GA[RND, pMPNN,pMPNN] setup, by embedding pMPNN into the mutation operator, we effectively extend the autoregressive sequence decoding model of pMPNN into that of a parallel, iterative sampling process as well, although "parallel" in the current context is more appropriately interpreted as a population-level parallelization.

A key property of diffusion models is that they can be used to integrate additional design objectives, whereby the gradient of any differentiable objective function or classifier can be added to the score function to bias the reverse diffusion process. The framework demonstrated in this work imposes no differentiability conditions. More importantly, however, in a diffusion model, these additional objective functions are collapsed into a scalar value, typically by taking a linear combination. This technique, called scalarization, is commonly used to reduce a multi-objective optimization problem into a single-objective one, but gradient-based optimization of such a linear combination can lead to incomplete recovery of the Pareto front, especially when the Pareto front is not convex [46]. In ensemble/population-based Monte Carlo methods such as NSGA-II, scalarization is unnecessary, because the Pareto front is naturally encoded in the population structure of the candidate solutions, but no equivalent structures exist for single-variable Monte Carlo methods such as diffusion models. Therefore, an interesting line of research is to investigate whether diffusion models can be extended into ensemble/ population-based Monte Carlo methods and be used to perform controlled generation explicitly conditioned on the Pareto front in an objective space. Ultimately, however, a more general approach is to construct a generative model that can be directly and efficiently conditioned on the free energy landscape defined in a reaction coordinate space; indeed, some recent progress has been made towards solving the inverse of this problem [39,84].

## Methods

### Structure preparation

The starting RfaH structures are taken from the Protein Data Bank. The RfaHα structure is taken from 5OND [85], after deleting the DNA and water molecules and the symmetric copy.

The RfaHα structure contains a missing loop at residues 98–117 that connects the N- and C-terminal domains, which is left unmodeled. The RfaHβ structure is taken from the first model of 2LCL [56]. In 2LCL, residues 1–96 are not resolved; furthermore, we delete residues 97–107 because of their high degree of conformational heterogeneity in the NMR structure ensemble, which makes such residues unsuitable for the structure-dependent design approach described in this work.

For the PapD benchmark, the PapD-PapE complex structure is taken from 1N0L [86] after deleting the water molecules, the PapD-PapK complex structure is taken from 1PDK [87] after deleting the water molecules, and the PapD homodimer structure is taken from 1QPP [88]. In 1QPP, one of the PapD chains contains missing residues at 123–124 and the other chain contains missing residues at 97–103; these missing residues were modeled by copying the corresponding region in the other chain into the missing region before Rosetta structure relax. In 1N0L, the PapE structure contains missing loop structures at residues 29–34, 128–136 and 73–77, which are left unmodeled because they are not located near the redesigned PapD-PapE interface.

For the CaM benchmark, the 14 input structures and the modifications made to them are described in Table 1. The canonical binding mode annotation used in S19A Fig is taken from [74]. All CaM structures used in this work are resolved in buffer conditions containing $Ca^{2+}$,

**Table 1. Structural models of CaM.**

| PDB ID | Description | Modifications |
|---|---|---|
| 1CFD [89] | apo-CaM | N/A |
| 1CFF [90] | CaM in complex with the peptide C20W representing plasma membrane calcium ATPase 4 (PMCA4) | Take the fourth frame from the NMR ensemble, because of its distinct conformational state in between a fully collapsed and fully extended state. |
| 1CKK [91] | CaM in complex with a peptide of $Ca^{2+}$/CaM-dependent protein kinase kinase (CaMKK) | Take the first frame from the NMR ensemble. |
| 1CLL [92] | CaM with no binding partner | N/A |
| 1CM1 [93] | CaM in complex with a peptide fragment of the CaM binding domain of calmodulin-dependent protein kinase IIα (CaMKIIα) | N/A |
| 1G4Y [94] | CaM in complex with the CaM-binding domain (CaMBD) of the small-conductance $Ca^{2+}$-activated $K^+$ channel subunit SK2 | N/A |
| 1NIW [95] | CaM in complex with a peptide fragment of the CaM-binding region of the endothelial nitric oxide synthase (eNOS) | N/A |
| 1NWD [96] | CaM in complex with two C-terminal peptides of glutamate decarboxylase (GAD) | Take the first frame from the NMR ensemble. |
| 2F2P [80] | CaM in complex with calcineurin peptides | N/A |
| 2N8J [97] | CaM in complex with a peptide fragment of the CaM-binding region of eNOS | Take the first frame from the NMR ensemble. |
| 2WEL [98] | CaM in complex with the δ isoform of $Ca^{2+}$/calmodulin-dependent protein kinase II (CaMKIIδ) | Delete all CaMKIIδ residues before 293, which are distal to the CaM binding site. |
| 3EWT [99] | CaM in complex with a peptide of tumor necrosis factor receptor superfamily member 6 (TNFRSF6) | N/A |
| 3EWV [99] | CaM in complex with a peptide of tumor necrosis factor receptor superfamily member 16 (TNFRSF16) | N/A |
| 4DJC [100] | CaM in complex with the DIII-IV linker of the cardiac sodium channel $Na_V1.5$ | N/A |

except for 1CFD. Note that both 1NIW and 2N8J contain the same binding partner, but CaM shows different conformations in the two structures (Fig 1B) because the structures were solved under different $Ca^{2+}$ concentrations. Ions, water and other small molecule ligands are deleted from the structures, wherever applicable.

All starting structures are relaxed using a custom script with PyRosetta [101]. Specifically, a gentle CA position restraint is applied to the starting structures, which are then relaxed using FastRelax [102,103] in dual space with the MonomerDesign2019 script and the beta_nov16_cart scoring function [104]. For each starting structure, 1000 relaxed structures are generated, and the structure with the lowest total score is chosen for subsequent simulations. Note that lowest-energy relaxed structures are used as inputs to all pMPNN and AlphaFold2 function calls; the input structures are not updated based on any further designed sequences or predicted structures.

Structural visualization is performed with PyMol (version 2.5.4). Relative solvent accessibility calculation on the relaxed structures is performed using the GETAREA webserver [105].

We note here that many of the structures used in this work to define alternative conformational states of a given protein are of different chain lengths, often due to missing residues at the N- and/or C-terminus. This does not preclude the application of pMPNN, as the only requirement for multistate design with pMPNN is that the equivalent positions to be tied together from the different states are present in the structures of these states.

The fact that some residues are present in only a subset of the structures for a given protein do have two consequences. First, the local chemical environment around some designable positions close to the undesigned missing residues will not reflect that of the full-length protein, which may lead to biases in native sequence recovery whenever a structure-based metric is involved. However, the fact that such biases, if present, are constant in all design simulation setups suggest that the relative difference in native sequence recovery and other performance metrics among the setups can be meaningfully attributed to the differences in the mutation operator and/or the objective functions.

Second, due to the different chain lengths, some of the structure-based metrics computed for one state of a protein may not be directly comparable to those computed for another. For example, the apparent bias of AF2Rank against the RfaHα state in the pMPNN-SD designs (Fig 2, last row) is an artifact, due to the fact that the AF2Rank score is calculated for the full protein structure of each state, but the N-terminal domain is absent in the structure of the RfaHβ state. As a result, the AF2Rank composite score of the RfaHα state has a higher lower bound than that of the RfaHβ state, and a single-state design simulation that optimizes the RfaHα state (at the expense of the RfaHβ state) will therefore have a lower hypervolume. As such, these structure-based metrics should only be compared within each state, as is done in this work, rather than between the states.

## Designable positions

For RfaH, the designable positions are the C-terminal domain residues 119 to 154. For PapD, the designable positions are the interface residues 1, 3, 4, 5, 6, 7, 8, 31, 91, 104, 105, 106, 107, 108, 109, 110, 112, 152, 154, 163, 164, 166, 170, 194, and 200 (taken from [75]). For CaM, the designable positions are residues 6–145, except for residues 20, 22, 24, 31, 56, 58, 60, 67, 93, 95, 97, 104, 129, 131, 133, and 140, whose sidechains are involved in $Ca^{2+}$ binding; residues before 6 and after 145 are excluded because they are not present in all CaM structures.

## ProteinMPNN

The code for ProteinMPNN (pMPNN) [20] (version 1.0.1) was accessed via GitHub from https://github.com/dauparas/ProteinMPNN. In this work, we perform sequence design with

pMPNN-AD and pMPNN-SD using the vanilla model weights and the default weight of 1.0 for each state. When pMPNN-SD is used as an objective function, the negative log likelihood score of a state is computed from all residue positions and averaged over 5 repeats (–score_only, –num_seq_per_target 5). Unless otherwise specified, all pMPNN runs are performed at a temperature of 0.3, which is chosen to balance native sequence recovery and sequence diversity; note that the pMPNN-SD log likelihood scores are temperature-independent.

To perform multistate design with pMPNN-AD, we input a combined PDB file containing the structures for all the states of the system, such that the pairwise centroid distance $r_{ij}$ between the structures of state $i$ and $j$ satisfies

$$r_{ij} \geq 2 \max(r_{i,\max},\ r_{j,\max}) + r_{\min}$$

Where $r_{i,\max}$ is the maximum distance between the centroid of structure $i$ and the set of all CA atoms in the structure, and $r_{\min}$ is a minimum distance set to 24 Å. This distance is chosen such that the backbone atoms from the structures of different states will have a shortest distance longer than the range of distances that can be encoded as edge features using the radial basis functions. The corresponding designable positions in the different states of the protein are then tied together during sequence decoding.

When the sequence encoded in the input structure does not match the sequence of a design candidate, as is almost always the case during genetic algorithm simulations, the sequence in the JSON file generated from parsing the input PDB file is updated before feeding to pMPNN.

## Genetic algorithm

In this work, we employ NSGA-II [55] as the primary evolutionary multiobjective optimization framework, through its implementation in the pymoo package [106] (version 0.6.0.1). Briefly, NSGA-II is an iterative algorithm, and, at each iteration, the population is doubled by the application of binary tournament selection, a crossover operator, and a mutation operator. Then, the new, combined population is partitioned by repeated non-dominated sorting into successive Pareto fronts. Lastly, the next generation is populated through elitist selection, whereby individuals from successive Pareto fronts are added to the next generation until the population size limit is reached, with a crowding distance sorting mechanism (in the objective space) as a tiebreaker if not all individuals on the last admitted front can be added. A detailed description and analysis of NSGA-II can be found in [55], and a broader survey of evolutionary multiobjective optimization algorithms can be found in [46].

With NSGA-II, we employ the standard binary tournament selection, $n$-point crossover operator, and a custom-constructed mutation operator to generate new candidates, with the crossover operator and the mutation operator applied with a probability of 0.9 and 1.0, respectively. For the $n$-point crossover operator, $n = 2$ unless otherwise specified. The custom mutation operator provides two methods for picking which subset of residue positions to redesign:

1. "Random" (or "RND"): each designable position is considered for redesign with a probability controlled by the mutation rate ($\mu$); if no position is chosen after this procedure, a single designable position is picked randomly. Because this method essentially generates a sequence of Bernoulli trials with a success probability of $\mu$, the number of design positions selected using this method follows a binomial distribution $\text{Bin}(n, \mu)$, where $n$ is the total number of designable positions.

2. "ESM": a random polypeptide chain containing designable positions is chosen first, and then the residue at each position in the chain is scored by ESM-1v [27] in a single pass with no masking. The designable positions are ranked from the lowest to the highest scores; the

first $k$ positions with the lowest scores are selected for redesign, where $k$ is either 1 or a random number drawn from a binomial distribution $\text{Bin}(n, \mu)$, whichever is larger; here, $n$ is the total number of designable positions.

Once a subset of design positions is selected, they are mutated through either uniform random sampling from the set of standard amino acids, or through pMPNN-AD. Lastly, the mutated sequences are scored using the pMPNN-SD negative log likelihood scores, the ESM-1v log likelihood scores, and/or the AF2Rank composite scores for each state; note that the sign for each objective function is chosen such that the simulation can be framed as a minimization problem. To accelerate the simulation, options are implemented to parallelize the evaluation of the mutation operator and the objective functions through either MPI (using mpi4py [107]) or a job scheduler.

In this work, we generate 100 sequences for each pMPNN-AD or pMPNN-SD benchmark condition, and perform simulations over 50 iterations with a population size of 100 for each genetic algorithm benchmark setup. Note that for all genetic algorithm benchmark simulations, the sequence at the designable positions are fully randomized in the initial population.

A detailed flowchart illustrating the GA[ESM,pMPNN,AF2Rank] setup is shown in S1 Fig. For the CaM sequence design problem, NSGA-III [108] is used in place of NSGA-II. NSGA-III is an extension of NSGA-II intended for higher-dimensional optimization problems, which prioritizes sampling near a set of user-specified reference directions. In this work, the reference directions are generated using a method based on Riesz s-energy [109], implemented in pymoo, and the number of reference directions is chosen to be equal to the population size.

To investigate the reproducibility of sequence populations generated by the genetic algorithm, we repeat RfaH simulations for the GA[ESM,pMPNN,AF2Rank] setup with four different mutation rates using a different starting random seed, which controls both the initial sequence population generated from randomized WT designable sequence and the random states of all genetic operators. Comparison of the two sets of simulation results suggest a high degree of reproducibility (S21 Fig).

## AF2Rank

The AF2Rank composite score is used in this work as a folding propensity metric. The method is described in [59] and the code was accessed from https://github.com/jproney/AF2Rank. Briefly, the sequence(s) for each state is provided to AlphaFold2 [14] via the ColabDesign package (version 1.1.1; code was accessed from https://github.com/sokrypton/ColabDesign); here, a state can include binding partners, even if they are not redesigned, and residues missing from the structure of the state are trimmed from the input. AlphaFold2 is called with 1 recycle using the model_1_ptm parameter set (or model_1_multimer_v3 [58] if more than one polypeptide chain is present). The structure for the state is provided as a template, after replacing the template sequence with gap tokens, deleting the sidechain atoms, and imputing the CB positions for glycine residues. No multiple sequence alignment is provided. A final composite score is defined as the product of average AlphaFold2 pLDDT (scaled from 0 to 1), AlphaFold2 pTM, and a TM-score [110] between the template and AlphaFold2 predicted structure. The AF2Rank composite score is therefore bounded between [0, 1], with a higher value being indicative of greater folding propensity.

Recently, several works have suggested that, by manipulating the input multiple sequence alignment, AlphaFold2 is sometimes able to predict alternative protein conformations [111–113]. We do not consider these techniques in this work, because their need for evolutionary information may restrict their applicability in a protein design pipeline where no such information is available.

### ESM-1v

The ESM-1v pre-trained protein language model [27] (esm1v_t33_650M_UR90S_1) is used in this work as a mutational effect predictor. Through a single forward pass with no masking, the model is used either to rank residue positions, or the per-position score is averaged to give the (pseudo)likelihood score for a sequence. It has been shown that this scoring method gives results that are highly correlated with the masked marginal probabilities generated by sequential masking of single positions, while only requiring a single pass of the model [114]. All calculations involving ESM-1v are done using the pgen package (version 0.2.3) [115,114], which is available at https://github.com/seanrjohnson/protein_gibbs_sampler.

### Hypervolume

Hypervolume is a standard metric used to assess the area/volume of the objective space, with respect to a reference point, covered by the approximate Pareto front generated using a genetic algorithm [63]. Therefore, the farther away the solutions are from the reference point, and the wider they are spread out over the approximate Pareto front, the higher the hypervolume. In this work, we use the origin as the reference point to measure hypervolume in the (negative) AF2Rank composite score subspace, and, somewhat arbitrarily, (4, 4, . . ., 4) (i.e., 4 repeated over the dimension of the objective space) as the reference point to measure hypervolume in the pMPNN-SD (negative) log likelihood score subspace.

For the RfaH and PapD benchmarks, the hypervolumes are calculated exactly, using pymoo. Because the computational cost of the exact method scales exponentially with objective space dimension, it is prohibitively expensive for higher-dimensional systems such as CaM. For CaM, an approximate hypervolume is computed instead using the Bringmann-Friedrich approximation scheme [116], implemented in the pygmoo package [117] (version 2.19.5), with a relative accuracy of 0.1 and probability of error of 0.05.

### Sequence analysis

Native sequence recovery is defined as the fraction of designable positions where the redesigned sequence is identical to the WT sequence. We report the population average native sequence recovery as a performance metric.

(Normalized) sequence similarity is calculated with the BLOSUM62 substitution matrix [118] over the designable positions using Biopython [119]; the per-position scores are averaged to produce the sequence similarity metric. The pairwise sequence similarity matrix is used as a precomputed affinity matrix for Laplacian Eigenmaps dimensionality reduction [65] using scikit-learn (version 1.1.3) with the default settings [120].

Principal component analyses of sequence distributions in the objective spaces (Figs 5 and S19) are performed using scikit-learn with the default settings. All single-linkage hierarchical clustering analyses (S19 and S20 Figs) are performed using scipy (version 1.9.3) with the default settings [121,122].

Visualizations of multiple sequence alignments are generated using Logomaker [123]; the color scheme follows that of [124].

### Comparison to NusG-like sequences

We compare the redesigned RfaH sequences to a database of NusG-like sequences that have been annotated by [72] as either foldswitching or non-foldswitching proteins.

We first filter out sequences in the NusG database where the foldswitching prediction confidence is low, and then randomly sample 1,000 each of foldswitching and non-foldswitching

sequences, out of the 15,195 sequences in total in the database, for further analysis. For each selected sequence, a new C-terminal domain sequence is defined, because [72] uses a different definition of C-terminal domain than the RfaH designable positions defined in this work. To do so, the full sequences are downloaded (in December 2023) from UniProtKB [125] using their UniProt IDs, and aligned with the *E. coli* RfaH sequence using Clustal Omega [126,127]; for each aligned sequence, residue positions that do not correspond to *E. coli* RfaH residues 119 to 154 are discarded, and the aligned sequence itself is discarded if its new C-terminal domain sequence contains gap(s). In the end, this results in a reduced set of 645 high-confidence, fully alignable, foldswitching NusG-like sequences, and 956 non-foldswitching ones.

The distance between a redesigned RfaH sequence and the reduced set of (non-)foldswitching sequences is defined as the 95% percentile of normalized BLOSUM62 sequence similarity between the redesigned RfaH sequence and each sequence in the reduced set.

## Structural modeling of redesigned sequences

Representative sequences from the RfaH, PapD, and CaM design simulations were chosen for structural analysis. For RfaH, the sequence closest to the GA[pMPNN] consensus sequence is chosen to illustrate the redesigned surface charge patterns (S12 Fig) and key mutations at F130 and S139 (S13 Fig) in GA[pMPNN]. For PapD, the sequence closest to the consensus sequence in the subpopulation of pMPNN-AD sequences that have the L107K mutation is chosen to understand the energetic effect of L107K mutation on the PapD binding interfaces (S16 Fig). For CaM, the sequence closest to the consensus sequence of GA[pMPNN] is chosen to assess the effect of additional negative charges on the CaM binding interfaces (S18 Fig). In all three cases, sequence distance is measured in terms of percentage sequence identity.

For the chosen sequences, their mutations are modeled structurally using a custom PyRosetta script implementing a method similar to [128]: for residues within a 10 Å distance from any of the mutated positions (as measured by CB-CB distance, or in the case of glycine, CA is used in place of CB atoms), a gentle CA constraint is applied, and these residues are allowed to be repacked and energy-minimized; for residues outside of this radius, a strong CA constraint is applied, and repacking and energy minimization is disabled. The mutations are introduced into the WT structure using the MutateResidue mover, and the mutated structure is relaxed using FastRelax in Cartesian space with the MonomerDesign2019 script and the beta_nov16_-cart scoring function.

For RfaH and PapD, this protocol is repeated 10 times for each state, and the mutated structures with the lowest energy are chosen for further visualization and analysis.

For the CaM interface analysis, this protocol is repeated 100 times for each state; in addition, this protocol is repeated another 100 times for each state, but without actually introducing the mutations, to produce new WT structures that have been relaxed using the same method as the mutant structures. These pairs of mutated and WT structures are then used in the interface analysis. The interface $\Delta\Delta E$ between CaM and its binding partners is defined in a way similar to that used in the Interface Analyzer in Rosetta [129]:

$$\Delta\Delta E = (E_{\text{mut}}[\text{C}:\text{P}] - E_{\text{mut}}[\text{C}] - E_{\text{mut}}[\text{P}]) - (E_{\text{WT}}[\text{C}:\text{P}] - E_{\text{WT}}[\text{C}] - E_{\text{WT}}[\text{P}])$$

where $E$ is the Rosetta total energy, $E_{\text{mut}}$ is the energy for structures containing the mutations, and $E_{\text{WT}}$ is the energy for structures without the mutations; "C:P" refers to structures containing both CaM and its binding partner, "C" refers to structures where the binding partner has been deleted, and "P" refers to structures where CaM has been deleted. To specifically quantify the electrostatic contribution to $\Delta\Delta E$, a separate quantity, $\Delta\Delta E_{\text{elec}}$ is defined using a similar formula, but instead of Rosetta total energy, only the fa_elec and fa_intra_elec terms are used.

## Supporting information

**S1 Fig. GA[ESM,pMPNN,AF2Rank] algorithm flowchart.** Elements of the algorithm typically found in genetic algorithms are shown in green, methods specific to Non-dominated Sorting Genetic Algorithm II (NSGA-II) are highlighted in yellow, the machine-learning models are highlighted in blue, the modifications to the mutation operator proposed in this work are highlighted in purple, and problem-specific elements are highlighted in red. The hyperparameters to the genetic algorithm and their values used in this work are shown with underlines (for the crossover and mutation operators, $p$ is the probability that the operator is applied). See Methods for additional details.
(PNG)

**S2 Fig. RfaH genetic algorithm benchmark simulations at additional mutation rates.** Refer to Fig 2 legend for more details.
(PNG)

**S3 Fig. Benchmark of the n-point crossover operator using RfaH.** Five sets of simulations are performed with the GA[ESM,pMPNN,AF2Rank;$\mu = 0.3$] setup, while varying the number of crossover points. For the uniform crossover operator, the residue at each designable position is chosen from either parental sequences with equal probability; effectively, for $L$ designable positions, the uniform crossover operator is analogous to the $(L-1)$-point crossover operator.
(PNG)

**S4 Fig. Random resetting operator results in RfaH designs with high sequence entropy.** Each panel shows the sequence profiles from the last iteration of the GA[RND,RND,pMPNN] (left column) or the GA[RND,RND,AF2Rank] (right column) setup at a different mutation rate (rows). See also S5 Fig for the cumulative distribution functions of the per-position sequence entropy.
(PNG)

**S5 Fig. RfaH sequence entropy distributions.** Each panel shows the empirical cumulative distribution function (eCDF) of per-position sequence entropy (base $e$) for the last iteration population generated with two genetic algorithm setups. Each row represents a mutation rate ($\mu$), and each column represents a mutation operator configuration ("+ESM" indicates that ESM-1v is used as a third objective function); the curves in green indicate that pMPNN-SD log likelihood is used as the objective functions, and the curves in purple indicate that AF2Rank composite score is used as the objective functions. The gray curves are calculated from the pMPNN-AD reference population. The black dashed lines represent ln 20, the entropy of a uniform distribution over the 20 standard amino acid types, which is the theoretical maximum entropy for any discrete distributions that can take on 20 values. See Fig 2 legend for details on the abbreviations used here.
(PNG)

**S6 Fig. AF2Rank objective space RfaH sequence distribution for GA[RND,RND, AF2Rank].** Each panel shows the distribution of the last iteration sequences generated using GA[RND,RND,AF2Rank] at a different mutation rate (purple points), compared to the WT sequence (black star), in the AF2Rank composite score objective space.
(PNG)

**S7 Fig. GA[RND,pMPNN,pMPNN] produces Pareto-optimal sequences that separate from the pMPNN-AD sequences in the RfaH objective space.** Each panel shows the

distribution of the last iteration sequences generated using GA[RND,pMPNN,pMPNN] at a different mutation rate (green), compared to the sequences generated from pMPNN-AD (gray), in the pMPNN-SD log likelihood objective space.
(PNG)

**S8 Fig. Using ESM-1v as an objective function biases sampling towards RfaHβ-like sequences with lower native sequence recovery.** Each panel shows the distribution of the population of the pMPNN-SD sequences and the last iteration population from a GA setup (specified in the title; all with mutation rate 0.3) in a two-dimensional embedding of the sequence space generated using Laplace eigenmaps, as in Fig 3. The sequences are colored by native sequence recovery, except for the WT sequence shown as the black stars; the pMPNN-SD sequences for the RfaHα and RfaHβ states are shown using "$\alpha$" and "$\beta$" as markers.
(PNG)

**S9 Fig. Pairwise correlation analysis of RfaH sequence design performance metrics.** The sequences shown are designed with pMPNN-AD (gray), GA[ESM,pMPNN,pMPNN;$\mu = 0.3$] (green; abbreviated as GA[pMPNN]), and GA[ESM,pMPNN,AF2Rank;$\mu = 0.3$] (purple; abbreviated as GA[AF2Rank]). All GA sequences in this figure refer to the final iteration sequence populations.
(PNG)

**S10 Fig. AF2Rank is sensitive to the differential sequence preferences of the two RfaH states.** The logo plots for the final iteration sequences designed by GA[ESM,pMPNN,pMPNN; $\mu = 0.3$] (i.e., GA[pMPNN]), which are partitioned here into three groups: "RfaHα-like", which has bad AF2Rank[RfaHβ] scores ($< 0.6$); "RfaHβ-like", which has bad AF2Rank [RfaHα] scores ($< 0.3$); and "mode", which consists of the rest of the sequences. See Fig 3A (third panel) for the distribution of these sequences in the AF2Rank objective space. The logo plots for the RfaHα and RfaHβ single-state design (SD) sequences are also shown for comparison. The positions highlighted in gray shading indicate major differences in the recovered residue types among the three subgroups of GA[pMPNN] sequences.
(PNG)

**S11 Fig. Post hoc non-dominated sorting does not lead to significant sequence entropy reduction for RfaH.** Non-dominated sorting of the 100 pMPNN-AD sequences is performed in the AF2Rank (top row) and pMPNN-SD (bottom row) objective spaces, and the sorting results are shown on the left column. The empirical cumulative distribution functions (eCDF) of the sequence entropies for the sorted, Pareto-optimal solutions are shown on the right column as colored continuous curves. For comparison, sequence entropies are also calculated for a random subsample of the pMPNN-AD sequences of the same size as the Pareto front; this procedure is repeated 100 times and the cumulative distributions of all subsampled sequence entropies is shown as the gray dashed curves. A similar random subsampling method is applied to the GA[AF2Rank] and GA[pMPNN] sequences, and the results are shown as colored dashed curves.
(PNG)

**S12 Fig. GA[pMPNN] introduces new surface charged residues and salt bridges in RfaH.** The structure models of the WT (left) and a redesigned (right) sequence chosen from GA [pMPNN] in the RfaHα (top) and RfaHβ (bottom) states are shown in cartoon representation. The sidechain conformations at the positions 127, 137, 140, 147, 152, and any undesigned positions that form salt bridge interactions with these residues, are highlighted in green stick

representation; the salt bridge interactions are represented as dashed yellow lines. See Methods for how the redesigned sequence is chosen and how its structural model is generated.
(PNG)

**S13 Fig. Mutations at RfaH F130 and S139.** The local structural environment around F130I in the RfaHα state is shown in A, and the local environment around S139V in the RfaHβ and RfaHα states are shown in B and C, respectively. The backbone of the redesigned structure is shown in cartoon representation, while the sidechains of the WT and redesigned residues are shown in white and green stick representation, respectively. The hydrogen bonding interactions are shown with dashed lines, with those in the WT structure in white and those in the redesigned structure in yellow. The redesigned sequence and its structural models shown here are the same as that in S12 Fig.
(PNG)

**S14 Fig. Comparison of GA[pMPNN] and GA[AF2Rank] to low temperature pMPNN-AD RfaH sequences.** A. Progression of GA[ESM,pMPNN,pMPNN] and GA[ESM,pMPNN, AF2Rank] simulations, reproduced from Fig 2. The dashed black lines represent the population averages from the pMPNN-AD sequences at $T = 0.1$, while the dotted gray line represents averages at $T = 0.3$, as per Fig 2. B. Distribution of pMPNN-SD, pMPNN-AD, and GA sequences, reproduced from Fig 3A, but with the pMPNN-AD population from the $T = 0.1$ simulation. C. Logo plots for sequences generated with pMPNN-AD at $T = 0.1$ (top) and $T = 0.3$ (bottom). The residue positions highlighted in gray shading show reduced or no native sequence recovery at $T = 0.1$ compared to $T = 0.3$.
(PNG)

**S15 Fig. Effect of state weights on RfaH pMPNN-AD sequence decoding.** For F123, F130, S139, M140, N144, and N147 (i.e., positions identified in Fig 3C in the "native recovery" category), we perform a single-position sequence design simulation with pMPNN-AD to extract the raw logits (not scaled by temperature or normalized by softmax) of all 20 standard residue types for the two RfaH states (first and third columns). The dashed gray lines represent the $y = x$ diagonal lines. Then, we ask how the probability assigned to each residue type (at temperature 0.3) changes as a function of the relative weight for the RfaHα state (second and fourth columns). If a residue type is ever assigned a probability $> 0.05$ as a function of state weight at a position, then the residue is shown in color on the panels associated with the position; otherwise the residue is shown in black. Note that the relative proportions of decoded residues using pMPNN-AD shown in Fig 3 will not necessarily match those at relative weight 0.5 shown here, because in practice sequence decoding at any given position is usually not performed with the full WT sequence context.
(PNG)

**S16 Fig. PapD sequences redesigned by pMPNN-AD are enriched in basic residues at L107.** A. Logo plots for the sequences designed using pMPNN-AD (abbreviated as AD) (top two rows), the last iteration of GA[ESM,pMPNN,pMPNN;$\mu = 0.3$] (abbreviated as GA [pMPNN]) (third row) and the last iteration of GA[ESM,pMPNN,AF2Rank;$\mu = 0.3$] (abbreviated as GA[AF2Rank] (last row). For pMPNN-AD, the redesigned sequences are split into two subpopulations, depending on the position of the sequences in the AF2Rank composite score objective space (Fig 5B, third panel): sequences that reside in the upper half of the principal component space (defined as having a second principal component (PC2) value greater than 0.2) are shown in the top row, and sequences that reside in the lower half are shown in the second row. The residue distributions at L107 are highlighted in gray. B. Structural analysis of L107K. A sequence from pMPNN-AD containing the L107K mutation is selected and used to

generate the structural models shown here (see Methods). In each panel, PapD is shown in green cartoon representation, and its binding partner is colored in orange; the binding partners are, from left to right: PapE, PapK, and PapD. All residue sidechains containing atoms within 4 Å of the L107K sidechain are represented in stick models, and hydrogen bonds within the neighborhood are represented by dotted yellow lines; the sidechains from the WT structures are colored in white, while those from the redesigned structural model are colored in the same way as the backbone cartoon representation. In each panel, E92 is labeled, and residues that form polar and charged interactions with L107K are also labeled. In particular, while E92 forms salt bridge interactions with L107K at the PapD–PapD interface, the geometry at the PapD–PapE and PapD–PapK interfaces do not appear to allow for direct salt bridge interactions between E92 and L107K.
(PNG)

**S17 Fig. CaM sequence logo.** Logo plot for redesigned CaM sequences from pMPNN-AD (abbreviated as AD), the last iteration of GA[ESM,pMPNN,pMPNN;μ = 0.3] (abbreviated as GA[pMPNN]) and the last iteration of GA[ESM,pMPNN,AF2Rank;μ = 0.3] (abbreviated as GA[AF2]). Due to the length of the designable sequence, the logo plot is split into four residue blocks, roughly corresponding to the four EF hands of CaM: EF1 (residues 6–39), EF2 (residues 40–75), EF3 (residues 76–112), and EF4 (residues 113–145); residues within the 6–145 range that are not redesigned (remain wild-type) are not depicted in the Figure (see Methods).
(PNG)

**S18 Fig. Charge content of the redesigned CaM sequences.** A. Empirical cumulative distribution functions (eCDF) of net charge count for sequences designed using pMPNN-AD (abbreviated as AD), the last iteration of GA[ESM,pMPNN,pMPNN;μ = 0.3] (abbreviated as GA[pMPNN]) and the last iteration of GA[ESM,pMPNN,AF2Rank;μ = 0.3] (abbreviated as GA[AF2Rank]); the WT net charge count is shown as the black vertical line. The net charge count is defined as the number of K and R, minus the number of D and E, in the designable positions. B. Interface energetics for a representative sequence designed by GA[pMPNN]. See Methods on how the sequence is chosen and details on the calculation of interface energetics. The net charge count of this chosen sequence is –29, and only CaM states with a binding partner are included in this analysis. REU stands for Rosetta energy unit. C. Distributions of charged residues in the sequences redesigned by pMPNN-AD, GA[pMPNN], and GA[AF2Rank] (same sequences and color scheme as panel A). The net charge fraction at each position is defined as the net charge count at that position, summed over the design population, and divided by the size of the design population. Positions with a K or R as the WT residues are represented by a blue dot, and positions with a D or E as the WT residues are represented by a red dot. The central portion of the linker region (residues 74–84) enriched with charged residues is highlighted in light yellow. The residues not redesigned are represented as gaps in the traces.
(PNG)

**S19 Fig. The CaM objective spaces have low effective dimensionalities.** A. Pairwise TM-scores for the CaM structures used for design. The TM-scores are calculated using residues 6–145 of CaM only. The pairwise TM-score matrix is arranged using a single-linkage hierarchical clustering to highlight the correlations within the set of CaM structures. Each CaM structure is described by two labels: the complex represented by the structure, followed by the PDB ID (column), and the binding mode represented by the structure (row). Unless labeled as "apo", the structures used were resolved in the presence of $Ca^{2+}$. For the binding mode labels: "1:2" and "2:2" indicates a 1:2 and 2:2 stoichiometric ratio between CaM and the binding

partner, respectively; "C-lobe" indicates that the binding partner is only associated with the C-terminal lobe of CaM; "ext." indicates that CaM is in an extended conformation; "d.s." indicates a domain-swapped CaM dimer; a label with numbers connected by dashes indicates that the structure is described by a canonical binding mode (the numbers represent the primary sequence distances of the anchoring hydrophobic residues on the binding partner). B. The eigenvalue spectra from principal component analysis (PCA) of the GA[ESM,pMPNN, pMPNN;μ = 0.3] (abbreviated as GA[pMPNN]) and GA[ESM,pMPNN,AF2Rank;μ = 0.3] (abbreviated as GA[AF2Rank]) simulations. The GA[pMPNN] PCA analysis is done over the last iteration sequences generated by GA[pMPNN], in the pMPNN-SD log likelihood score objective space, while the GA[AF2Rank] PCA analysis is done over the last iteration sequences generated by GA[AF2Rank], in the AF2Rank composite score objective space. The two outlier sequences in Fig 5D (second panel) are removed prior to the PCA analysis. For each PCA, the eigenvalues are divided by the sum of all eigenvalues to produce the explained variance ratio (i.e., fraction of the total variance in the objective space explained by each corresponding principal component). Note that the PCAs examined here are different from those used for visualization in Fig 5B and 5D, which are performed for all GA[pMPNN], GA[AF2Rank], and pMPNN-AD designed sequences in each objective space. C. Loading plots for the first two principal components (PC). The left panel shows the loading plot for the GA[pMPNN] PCA, and the right panel shows the loading plot for the GA[AF2Rank] PCA. Vectors representing CaM states with a canonical binding mode are highlighted in purple. Vectors with a length less than 0.03 are not labeled.
(PNG)

**S20 Fig. The CaM per-position pMPNN-SD logits are highly correlated.** A. The distribution of pairwise correlation coefficients for pMPNN-SD logits. The per-position pMPNN logit vectors are extracted as for RfaH (see S15 Fig). For each designable position, the Pearson's correlation coefficient is calculated for each pair of logit vectors (i.e., for each pair of CaM chains), and the correlation coefficients for all positions are pooled to generate the histogram. B. The distribution of cluster numbers as a function of correlation coefficient threshold. For each designable position, a single-linkage hierarchical clustering is performed on the Pearson's correlation coefficient matrix over the pMPNN logit vectors. The logit vectors are then clustered by cutting the resulting dendrogram at a given correlation coefficient threshold. The number of clusters per position, which we interpret as the per-position effective dimension, is pooled to generate the box plot. The box plot extends from the first to the third quartile of the distributions, with the median line shown in between; the fliers represent outliers of the distributions. The dashed line at 16 indicates the maximum possible dimensionality; note that this is higher than the 14 states used to define the design problem, because two of the states contain CaM dimers. C. Correlation between the per-position effective dimensionality (i.e., the number of clusters per position in panel B) and per-position native sequence recovery (nat. seq. recov.) From left to right, each panel represents the correlation for sequences designed using pMPNN-AD (abbreviated as AD), the last iteration of GA[ESM,pMPNN,pMPNN;μ = 0.3] (abbreviated as GA[pMPNN]) and the last iteration of GA[ESM,pMPNN,AF2Rank;μ = 0.3] (abbreviated as GA[AF2Rank]). The per-position native sequence recovery is computed as the number of designed sequences with the WT residue at a given position, divided by the sequence population size. The per-position effective dimensionality is computed using a correlation coefficient threshold of 0.8.
(PNG)

**S21 Fig. Effect of starting random seed on RfaH simulation results.** A. Comparison of convergence metrics for GA[ESM,pMPNN,AF2Rank] simulations at four mutation rates using

two different starting random seeds. See Fig 2 for more details on the metrics. The simulation results presented in the rest of this work is based on seed 1. B. Comparison of the last iteration sequence logos at the mutation rate $\mu = 0.3$ for the two random seeds. Positions with major differences in the recovered sequence profiles are highlighted in gray shading; in both simulations, the WT residue types are recovered at these positions, but the simulations differ in terms of the alternative residue types recovered at these positions.
(PNG)

## Author Contributions

**Conceptualization:** Lu Hong, Tanja Kortemme.

**Data curation:** Lu Hong.

**Formal analysis:** Lu Hong.

**Funding acquisition:** Tanja Kortemme.

**Investigation:** Lu Hong.

**Methodology:** Lu Hong.

**Resources:** Tanja Kortemme.

**Software:** Lu Hong.

**Supervision:** Tanja Kortemme.

**Validation:** Lu Hong.

**Visualization:** Lu Hong.

**Writing – original draft:** Lu Hong.

**Writing – review & editing:** Tanja Kortemme.

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
