## [Decision Letter · Decision Letter 0]

2 Apr 2024

Dear Prof. Kortemme,

Thank you very much for submitting your manuscript "An integrative approach to protein sequence design through multiobjective optimization" for consideration at PLOS Computational Biology.

As with all papers reviewed by the journal, your manuscript was reviewed by members of the editorial board and by several independent reviewers. In light of the reviews (below this email), we would like to invite the resubmission of a significantly-revised version that takes into account the reviewers' comments.

We cannot make any decision about publication until we have seen the revised manuscript and your response to the reviewers' comments. Your revised manuscript is also likely to be sent to reviewers for further evaluation.

Sincerely,

Alexey Onufriev

Academic Editor

PLOS Computational Biology

Arne Elofsson

Section Editor

PLOS Computational Biology

Reviewer's Responses to Questions

**Comments to the Authors:**

Reviewer #1: This manuscript describes methodological advances relevant to protein design. Specifically, it examines multi-objective optimization using recently introduced machine-learning approaches for sampling sequence space and evaluating models within a genetic algorithm framework. The application is a fold-switching protein that presents significant challenges for design via established protocols.

Overall, the analysis is interesting, but a main concern is that the methodology is tested only for one system, and it is unclear whether the findings have broader relevance for other systems. It seems that at least several different systems should be tested before conclusions with any level of significance can be obtained.

In terms of presentation, the paper is dense at many places and lacking clarity. Especially at the beginning it is difficult to understand what exactly was done. It would help to have a flowchart in addition to or instead of Figure 1 that clearly illustrates the protocol, especially with respect to what type of output (sequence, structure, score) is obtained based on what type of input with the different tools and how this data is used within the genetic algorithm optimization framework. It probably would also help to explicitly state in the introduction what the main benchmark goal is (native sequence recovery based on a given structure, or in this case a set of structures) as that may not be obvious to readers that are not intimately familiar with protein design method assessment.

Additional minor comments/questions:

Why were residues 97-107 deleted from the 2LCL structure and does it matter that the Hbeta and Halpha structures apparently have different amino acid lengths?

How sensitive is the protocol to the initial structural ensemble?

Reviewer #2: This manuscript investigates whether integrating different (deep learning) models and objective functions leads to improved computational protein sequence design. By leveraging a Genetic Algorithm as the optimization framework, AlphaFold2 and ProteinMPNN estimated confidence scores are used to define the objective space, and a mutation operator based on ESM-1v and ProteinMPNN to rank and then redesign the least favorable positions. Using the foldswitching protein RfaH as a case study, the authors demonstrate that the proposed evolutionary multiobjective optimization approach improves upon off-the-shelf application of ProteinMPNN. The manuscript is well-written and easy to follow. The experimental design is well-reasoned and justified. I particularly appreciate the in-depth analyses of the results presented and the thoughtful discussion.

I have couple of suggestions for the authors to consider:

1.) One advantage of the proposed approach over the existing Monte Carlo sampling methods to invert models such as AlphaFold2 for protein design is to that it minimizes the generation of “adversarial” sequences having poor biophysical properties that nevertheless optimize the objective functions. While such a prospect seems reasonable in theory since an adversarial sequence can be propagated only if it is overfitted to by all models, is it practically the case? Can the authors perform some additional experiments to empirically demonstrate that the proposed approach does in fact minimize the generation of adversarial sequences over some of the existing Monte Carlo sampling methods based on inverted AlphaFold2?

2.) The authors used a baseline custom mutation operator called “Random” (or “RND”) in which each designable position is considered for redesign with a probability controlled by the mutation rate (μ); if no position is chosen after this procedure, a single designable position is picked randomly. Can the authors slightly modify it to come up with an additional baseline operator which can be named “Random_Bin” (or “RND_BIN”) in which k positions randomly chosen are selected for redesign, where k is either 1 or a random number drawn from a binomial distribution Bin(n, μ), whichever is larger; here, n is the total number of designable positions. It may be interesting to include a comparison between “RND_BIN” and “ESM”.

**Have the authors made all data and (if applicable) computational code underlying the findings in their manuscript fully available?**

Reviewer #1: Yes

Reviewer #2: Yes

PLOS authors have the option to publish the peer review history of their article (what does this mean?). If published, this will include your full peer review and any attached files.

Reviewer #1: No

Reviewer #2: No
---

## [Decision Letter · Decision Letter 1]

25 Jun 2024

Dear Prof. Kortemme,

We are pleased to inform you that your manuscript 'An integrative approach to protein sequence design through multiobjective optimization' has been provisionally accepted for publication in PLOS Computational Biology.

Best regards,

Alexey Onufriev

Academic Editor

PLOS Computational Biology

Arne Elofsson

Section Editor

PLOS Computational Biology

Reviewer's Responses to Questions

**Comments to the Authors:**

Reviewer #1: The revised version fully addresses my concerns.

Reviewer #2: My comments have been addressed.

**Have the authors made all data and (if applicable) computational code underlying the findings in their manuscript fully available?**

Reviewer #1: Yes

Reviewer #2: None

PLOS authors have the option to publish the peer review history of their article (what does this mean?). If published, this will include your full peer review and any attached files.

Reviewer #1: No

Reviewer #2: No

---

## [Editor Report · Acceptance letter]

7 Jul 2024

PCOMPBIOL-D-24-00356R1 

An integrative approach to protein sequence design through multiobjective optimization

Dear Dr Kortemme,

I am pleased to inform you that your manuscript has been formally accepted for publication in PLOS Computational Biology. Your manuscript is now with our production department and you will be notified of the publication date in due course.

With kind regards,

Olena Szabo
